

# Past, present and future rainfall erosivity in Central Europe based on convection-permitting climate simulations

Magdalena Uber[1], Michael Haller[2], Christoph Brendel[2], Gudrun Hillebrand[1], Thomas Hoffmann[1]

[1]Department Fluvial Morphology, Sediment Dynamics and Management, Federal Institute of Hydrology, 56068 Koblenz, Germany
[2]Deutscher Wetterdienst, 63067 Offenbach am Main, Germany

*Correspondence to*: Magdalena Uber (uber@bafg.de)

**Abstract.** Heavy rainfall is the main driver of soil erosion by water which is a threat to soil and water resources across the globe. As a consequence of climate change, precipitation -and especially extreme precipitation- is increasing in a warmer world, leading to an increase in rainfall erosivity. However, conventional global climate models struggle to represent extreme rain events and cannot provide precipitation data at the high spatio-temporal resolution that is needed for an accurate estimation of future rainfall erosivity. Convection-permitting simulations (CPS) on the other hand, provide high-resolution precipitation data and a better representation of extreme rain events, but they are mostly limited to relatively small spatial extents and short time periods. Here we present for the first time rainfall erosivity and soil erosion scenarios in a large modelling domain such as Central Europe based on high-resolution CPS climate data generated with COSMO-CLM. We calculate rainfall erosivity for the past (1971-2000), present (2001-2019), near future (2031-2060) and far future (2071-2100) and apply the new data set in the soil erosion model WaTEM/SEDEM for the Elbe River basin. Our results showed that future increases in rainfall erosivity in Central Europe can be up to 84 % in the river basins of Central Europe. These increases are much higher than previously estimated based on regression with mean annual precipitation. In consequence, soil erosion and sediment delivery in the Elbe River basin are also increasing strongly. Locally, changes in erosion rates can be as high as 120 %. We conclude that despite remaining limitations, convection-permitting simulations have an enormous and to date unexploited potential for climate impact studies on soil erosion. Thus, the soil erosion modelling community should follow closely the recent and future advances in climate modelling to take advantage of new CPS for climate impact studies.

## 1 Introduction

Soil erosion by water is one of the main threats to soils worldwide (Amundson et al., 2015; Panagos et al., 2015b). It causes severe ecological and socio-economic problems such as ecosystem degradation (Bilotta and Brazier, 2008; Orgiazzi and Panagos, 2018; Mueller et al., 2020; Stefanidis et al., 2022), loss of fertile topsoil on agricultural land (Pimentel et al., 1995; Zhao et al., 2013; Sartori et al., 2019), channel and reservoir siltation (Wisser et al., 2013; Kondolf et al., 2014) and nutrient and contaminant transport to water bodies (Owens et al., 2005; Ciszewski and Grygar, 2016). Heavy rainfall is the main driving force of soil erosion by water. It acts via the detachment of soil particles by raindrop impact or shear forces of overland flow





and subsequent transport of soil particles with overland flow. Rainfall erosivity is defined as "the capability of rainfall to cause soil loss from hillslopes by water" (Nearing et al., 2017). It is most commonly expressed as the R-factor of the Universal Soil Loss Equation (USLE, Wischmeier and Smith, (1978)) and its revised versions RUSLE (Renard et al., 1993) and RUSLE2 (USDA Agricultural Research Service, 2008). The USLE, its derivates and models based on the USLE are the most widely
used soil erosion models (Borrelli et al., 2021). The USLE calculates average annual soil loss at a site from rainfall erosivity, soil erodibility, topography, crop management and control practices.

Rainfall erosivity is governed by rainfall kinetic energy, which depends itself on raindrop numbers, sizes and fall velocities (Wilken et al., 2018). As drop size distributions and fall velocity distributions are usually not available for long periods of time and large study sites, rainfall intensity is usually used as a proxy. Numerous kinetic energy - rainfall intensity relations exist
in the literature and are used in soil erosion modelling (Van Dijk et al., 2002; Wilken et al., 2018; Brychta et al., 2022). Thus, site-specific rainfall erosivity expressed as the USLE R-factor is commonly calculated from long lasting precipitation records from rain gauges. R-factor values depend strongly on the temporal resolution of the underlying precipitation data time series. R-factors decrease with decreasing resolution of the precipitation data because intensity peaks are reduced when precipitation is aggregated over longer time spans (Fischer et al., 2018). When high-resolution precipitation data is available only at a few
locations or limited time periods but low-resolution data (daily – annual) is available elsewhere (e.g. denser rain gauge networks, past reconstructions or future projections), so called low-resolution approaches can be applied (Brychta et al., 2022). This approach is based on empirical relations between rainfall erosivity calculated from high-resolution data and lower resolution rainfall amounts (usually monthly, seasonal or annual totals). Application of these approaches to calculate future changes of rainfall erosivity is not permitted if the frequency distribution of rainfall events changes, as expected under climate
change.

Erosion modelling usually requires contiguous data of rainfall erosivity which is highly variable in space (Auerswald et al., 2019a). This spatial variability is usually not represented by rain gauge networks, so spatially interpolated raster data are necessary. Gauge-adjusted radar rainfall data has a high potential for the estimation of highly resolved and contiguous rainfall erosivity maps (Fischer et al., 2018; Risal et al., 2018; Auerswald et al., 2019a; Kreklow et al., 2020). Where ground-based
radar data is not available, satellite-based gridded precipitation data sets can also be used to generate contiguous maps (Vrieling et al., 2010; Teng et al., 2017). However, their errors have to be assessed critically (Phinzi and Ngetar, 2019).

As a result of climate change, precipitation is likely to increase in a warmer world (Sun et al., 2007; Fowler et al., 2021). For Central Europe, a net increase of total precipitation is projected with a decrease in summer and an increase in winter (Brienen et al., 2020; Jacob et al., 2014). Furthermore, extreme precipitation is increasing in an intensified hydrologic regime under
climate change in many parts of the world as well as in Central Europe. This is due to the fact that the share of convective precipitation in total precipitation is increasing (Berg et al., 2013). Trends of an increase in the frequency and intensity of extreme precipitation have been observed since the beginning of the 20[th] century (Groisman et al., 2005; Alexander et al., 2006; Arnone et al., 2013; Kendon et al., 2014; Fischer and Knutti, 2016) and are expected to continue in the future (Allen and Ingram, 2002; Kharin et al., 2013; Scoccimarro et al., 2013; Westra et al., 2013; Westra et al., 2014; Kendon et al., 2017;



Fowler et al., 2021). Thus, rainfall erosivity and soil erosion are also observed to increase and expected to increase further (Nearing et al., 2004; Mueller and Pfister, 2011; Hanel et al., 2016a; Panagos et al., 2017; Panagos et al., 2022; Auerswald et al., 2019a; Auerswald et al., 2019b; Borrelli et al., 2020).

For climate impact studies on soil erosion, a common limitation is the lack of reliable high-resolution precipitation data for the future (Eekhout and De Vente, 2020). Projections of future precipitation from regional climate models in Europe are typically

available at a temporal resolution of one day and a spatial resolution of 0.11° (approx. 12 km) (e.g. Jacob et al., 2014). Thus, low-resolution approaches based on regression models that estimate future R-factors from monthly or annual precipitation are commonly applied (Eekhout and De Vente, 2020). Out of 68 climate impact assessment studies reviewed by Eekhout and De Vente (2020) only four used sub-daily precipitation data. In the review of 3030 soil erosion modelling studies by Borelli et al. (2021), 196 were identified to have the aim to model "climate change" or "Land use change and climate change" impacts.

Only 11 studies are quoted to use sub-daily precipitation data. The few studies that use hourly or sub-hourly future precipitation data mostly use either statistical downscaling of lower resolution data (Routschek et al., 2015; Wang et al., 2018) or artificially generated precipitation time series (e.g. Coulthard et al., 2012; Simonneaux et al., 2015). Strictly speaking, regression-based models applying monthly or annual precipitation are only valid for the time period for which these models are calibrated and lead to underestimations of the rainfall erosivity if extreme precipitation events increase with time, as suggested by many

climate change scenarios.

Only recently, the development of convection-permitting climate simulations (CPS) offers the possibility to model rain erosivity considering the effects of changing frequency of heavy precipitation that predominantly drives future soil erosion. Thus, CPS have an enormous and to date unexploited potential for the calculation of future rainfall erosivity. Usually, convection-permitting means that the standard parametrization of convective rainfall is switched off in the model setup,

allowing the model to simulate the precipitation explicitly in each grid cell. As the grid size of most CPS still ranges between 2 and 4 km, large deep convection cells are explicitly simulated, while smaller shallow convection still needs to be simulated using a parametrization. Despite this shortcoming, CPS provide an improved representation of extreme precipitation compared to climate models with parametrized deep convection. This is due to several improvements: the diurnal cycle is strongly improved (Ban et al., 2014; Prein et al., 2015; Haller et al., 2023), the return periods of extreme precipitation are better

represented (Rybka et al., 2022) and added value diagnostics have been applied for the comparison to coarse climate model data (Raffa et al., 2021; Haller et al., 2023). However, CPS also show some limitations. The simulations on the km-scale for regional domains are still time-consuming and they need a considerable amount of computing power. Compromises have to be made: either the covered time period is shortened or the model domain is restricted to the region of interest. Up to some years ago, only single CPS have been performed, covering only one future scenario. Thus, given the novelty of CPS, model

ensembles are not yet available for regional model domains, at the time scales needed for the robust estimation of rainfall erosivity (~ 20 years) or for several emission scenarios. Lately, the first-of-its-kind CPS ensembles have been created through combined efforts in the CPS community (Coppola et al., 2020; Ban et al., 2021). Even though these ensemble simulations do





not cover the long time periods needed for the estimation of rainfall erosivity yet, these flagship studies show promising results that suggest that in the future ensembles of CPS will be available for climate impact studies including studies on soil erosion.

The COSMO-CLM is a regional climate model that is used on horizontal scales from 1 to 50 km (Rockel et al., 2008). It is the climate version of the former operational forecast model COSMO of the German meteorological service (Deutscher Wetterdienst, DWD) and other European partners. COSMO-CLM is jointly maintained and developed by the CLM-Community but will soon be gradually replaced by the newly developed model ICON-CLM (Pham et al., 2021). In the framework of the project BMDV-Expertennetzwerk, convection-permitting climate simulations have been performed with

COSMO-CLM. Three time periods including one historical period (1971-2000, called CPS-hist) and two future periods (CPS-scen; near future: 2031-2060; far future: 2071-2100) were simulated by dynamically downscaling from global model data. Additionally, evaluation simulations were conducted with reanalysis data forcing for the time period 1971-2019 (CPS-eval). The data are published and usable for manifold analyses and impact model studies.

The improved representation of extreme precipitation in CPS compared to conventional convection-parametrized climate

models as well as the high spatio-temporal resolution of CPS is of great benefit for climate impact studies in soil erosion modelling (Chapman et al., 2021). Using CPS with high temporal resolution facilitates the direct calculation of the R-factor and avoids the application of regression equations between the R-factor and annual precipitation, which are established for past climates but may not be valid for future climate with different precipitation frequency and magnitude. To our knowledge, to date only one study (Chapman et al., 2021) assessed the impact of climate change on soil erosion using a convection-

permitting climate model. They used 15-min precipitation data from the pan-African model CP4A to calculate rainfall erosivity in Tanzania and Malawi for 8 years in the past and 8 years in the future. Their results suggested that convection-parameterized regional and global climate models might underestimate future rainfall erosivity while CPS represent observed storm characteristics better. Nonetheless, there are remaining limitations of CPS that hinder their use in soil erosion modelling:

i) limited spatial extent of most CPS. While regional and global convection-parameterized simulations cover the entire
globe, to date CPS are only available for some regions.

ii) the relatively short periods of time covered by CPS. Because of the high interannual variability of rainfall erosivity, long time series are required for robust estimates of long-term R-factors. Wischmeier et al. (1958) give a minimum of 20 years.

iii) the lack of model ensembles. While ensembles of regional or global climate models give more robust estimates of
the future climate than single ensemble members, ensembles of CPS are only being developed recently.

Covering an area of approx. 1.6 million km$^2$ on land and in total 109 years, the CPS performed at DWD with COSMO-CLM overcome the limitations i) and ii) for the first time and are thus a valuable source of precipitation data for the estimation of rainfall erosivity in Central Europe.

In this study, we calculated rainfall erosivity in Central Europe expressed as the USLE R-factor for the past (1971-2000),
present (2001-2019), near future (2031-2060) and far future (2071-2100) from convection-permitting climate model output.





We assessed changes in rainfall erosivity from the climate model output for a historical and future time period. Furthermore, a first application of the new rainfall erosivity maps is presented. To this end, changes in soil erosion and sediment delivery to the Elbe River are modelled in the USLE-based model WaTEM/SEDEM. Finally, we discuss the potential and limitations of using CPS for the calculation of rainfall erosivity. The main remaining limitation is the fact that to date ensembles of CPS do

not exist yet at time scales covering at least 20 years needed for robust rainfall erosivity estimations. In consequence, the uncertainty due to the choice of the model and the emission scenario cannot be assessed. To address this problem, we compare or results to the ones obtained from an ensemble of conventional regional climate models as well as to results from the literature. To our knowledge, this is the first test case that applies CPS for the calculation of rainfall erosivity and soil erosion covering national spatial scales and temporal scales in the order of 30 years.

## 140    2 Material and methods

### 2.1 COSMO-CLM

Convection-permitting simulations were conducted using the non-hydrostatic regional climate model COSMO-CLM. It shares almost all relevant modules with the weather forecast model COSMO (Doms et al., 2001), which has been the operational weather forecast model of DWD for more than a decade, before it was replaced by the ICON model (Giorgetta et al., 2018) in

recent times. COSMO-CLM, the climate version of COSMO, is optimized for long-term climate runs for more than 15 years (Rockel et al., 2008; Sorland et al., 2021). The general COSMO characteristics (e.g. physics) are documented in Steppeler et al. (2003). The COSMO-CLM is described in more detail in Böhm et al. (2006). COSMO-CLM is usable on different horizontal grid widths and has a typical vertical spacing of 50 layers in the troposphere and the lower stratosphere up to about 22 km. Subgrid-scale physical processes are parametrized, as they cannot be calculated explicitly. For grid spacings of less

than 4 km, the convection parametrization scheme for deep convection is turned off, while the shallow convection scheme remains turned on. In the COSMO-CLM standard parametrization for coarser grid resolutions, both parts are switched on.

The model domain of the CPS has 415 x 423 grid points and is centred over Germany. It includes large parts of neighbouring countries and therefore fully covers the contributing catchment areas of the major Central European rivers Rhine, Elbe, Oder and Upper Danube until Bratislava (Fig. 1). The grid resolution is 0.0275° ($\approx$ 3 km). The model uses the standard

parametrizations for turbulence and (shallow) convection as well as for the time integration.



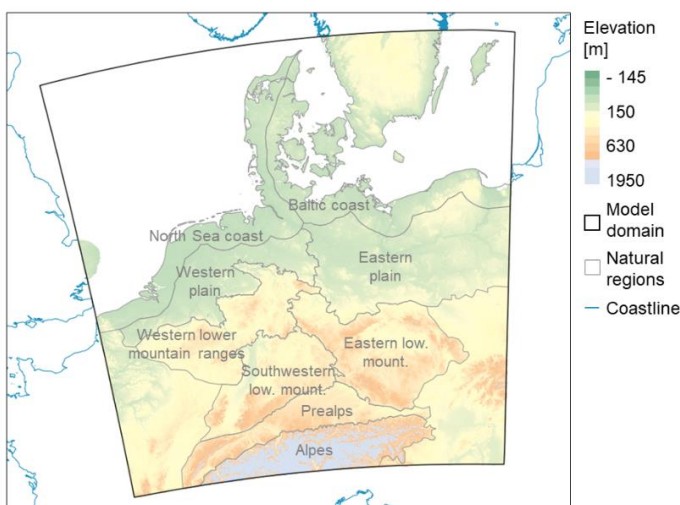

**Figure 1: Extent of the CPS model domain. Colours show the elevation (source: EU-DEM, Copernicus Land Monitoring Service, 2016). The natural regions outlined in grey were adapted from Bundesamt für Naturschutz (2017).**

For the projection simulations, three 30-year time slices have been selected, covering the years 1971-2000 (historical, CPS-
hist) and 2031-2060/2071-2100 (scenario, CPS-scen). CMIP5 global model data from the MIROC-MIROC5 (Watanabe et al., 2010) have been dynamically downscaled, applying the RCP8.5 scenario. The downscaling has been performed using an intermediate nesting step of 12 km.

The CPS evaluation simulation (CPS-eval) covering the time range of 1971–2019 is driven by ERA5 (Hersbach et al., 2020) for the years 1979 to 2019 and by ERA-40 reanalysis data (Uppala et al., 2005) for the years 1971 to 1978. ERA5 and ERA40
are reanalysis data that provide a comprehensive and coherent information of essential climate variables by assimilating additional various observational data to a model grid. The model system itself remains unchanged throughout the entire time period, resulting in a consistent approach to data assimilation and various parameterizations. For the ERA-40-driven time period, we used a two-fold nesting with a middle step at 0.11°, while for the ERA5-driven time period, a direct downscaling from 30 to 3 km was applied.

The model output consists of hourly data for the most important variables (e.g. temperature, precipitation, humidity and wind). It is available at https://esgf.dwd.de/projects/dwd-cps/ (accessed 10 February 2023) (Brienen et al., 2022; Haller et al., 2022a, b). The overall configuration of our simulation has been taken from the joint FPS-convection contribution in the CLM-community (Coppola et al., 2020). The hourly precipitation data that was further processed for this study was organized in five data sets, i.e. the projection simulations for the historical period, the near and far future (CPS-hist, CPS-scen-nf, CPS-scen-ff)
as well as the evaluation simulations for the historical period and the present (CPS-eval-hist, CPS-eval-present).


## 2.2 Calculation of rainfall erosivity

### 2.2.1 High temporal resolution approach

Following Wischmeier and Smith (1958; 1978) and Wischmeier (1959), the erosivity of an erosive rain event $R_e$ [N h$^{-1}$] is calculated as the product of maximum 30 min rain intensity $I_{max30}$ [mm h$^{-1}$] and kinetic energy $E_{kin}$ [kJ m$^{-2}$] of the rain event:


$$R_e = I_{max30} * E_{kin} \qquad\qquad (1)$$

An erosive rain event is defined as having a total precipitation (P) of at least 12.7 mm or a maximum 30 min rainfall intensity ($I_{max30}$) of at least 12.7 mm h$^{-1}$ and at least 6 h separate two erosive rain events. We use the classical equation by Wischmeier and Smith (1978) to calculate kinetic energy based on high-resolution rainfall data. Transferred to SI units, it calculates specific kinetic energy per millimeter rain depth, $e_{kin,i}$ in kJ m$^{-2}$ mm$^{-1}$ for every time increment during an erosive rainfall event as

follows (Rogler and Schwertmann, 1981):

$$e_{kin,i} = \begin{cases} 0 & for\ I < 0.05\ mm\ h^{-1} \\ (11.89 + 8.73 * \log_{10} I) * 10^{-3} & for\ 0.05\ mm\ h^{-1} \leq I < 76.2\ mm\ h^{-1} \\ 28.33 & for\ I \geq 76.2\ mm\ h^{-1} \end{cases} \qquad (2)$$

To obtain $E_{kin}$ for each event, $e_{kin,i}$ is multiplied with rain depth of each timestep and summed up for the entire rain event. Annual rainfall erosivity of a specific year is obtained by summing up $R_e$ of all erosive rain events in that year. The USLE R-factor is the long-term average of annual rainfall erosivity. R-factors are often given in the unit MJ mm ha$^{-1}$ h$^{-1}$ a$^{-1}$. To convert

rainfall erosivity as given here in N h$^{-1}$ a$^{-1}$ to MJ mm ha$^{-1}$ h$^{-1}$ a$^{-1}$, it has to be multiplied with a factor 10.

Here we calculated annual erosivity as well as long-term average annual erosivity for each one of the 175,545 grid points and for each of the five data sets (CPS-hist, CPS-scen-nf, CPS-scen-ff, CPS-eval-hist, CPS-eval-present). We used the command line suite Climate Data Operators (CDO, Schulzweida, 2022) and the library ncdf4 (Pierce, 2019) of the statistical software R to extract time series of 30 years (19 years for CPS-eval-present) for each grid cell and each data set and calculated rainfall

erosivity as described above. As the COSMO-CLM model output is available at a temporal resolution of 60 min, three adjustments were made as proposed by Fischer et al. (2018): (i) the rainfall intensity threshold of $I_{max30}$ to define an erosive rain event was lowered from 12.7 mm h$^{-1}$ to 5.8 mm h$^{-1}$, (ii) $I_{max30}$ in Eq. (1) was replaced by maximum 60 min rainfall erosivity $I_{max60}$ and (iii) a temporal scaling factor of 1.9 was applied to the R-factor for Germany to account for the reduction of intensity peaks with lower temporal resolution data. Here we did not apply a spatial scaling factor because it is unclear if such a

modification is necessary for climate model output.

We further assessed the seasonal distribution of erosivity by calculating the erosion index for each day of the year. The erosion index gives the contribution of each day to annual erosivity. The seasonal distribution of erosivity is important for soil erosion assessments, because of its interactions with seasonal changes of the crop cover. Briefly, high rainfall erosivity in months where vegetation cover is scarce (in Central Europe the winter months) is more severe than high rainfall erosivity during the

vegetation period (i.e. the summer months). The erosion index was calculated for each one out of 175,545 grid points and each





day of each year and averaged over all grid points and all 30 years in the three data sets from the projection simulations (CPS-hist, CPS-scen-nf, CPS-scen-ff).

### 2.2.2 Low temporal resolution approach

For comparison, we also calculated rainfall erosivity (R) for the past (1971-2000), near future (2031-2060) and far future
(2071-2100) from mean annual precipitation (MAP). Therefore, we used the empirical regression equation

$$R \ [N \ h^{-1} a^{-1}] = 0.0788 * MAP \ [mm] - 2.82 \tag{3}$$

from the German norm DIN 19708 (Din-Normenausschuss Wasserwesen, 2017), which was derived from regression analysis of R-factor values calculated based on Eq. (1) and annual precipitation sums for the time period from the 1960s to the 1980s in Germany. We used the median of the MAP of a climate model ensemble consisting of 21 members that were run with the
emission scenario RCP 8.5. The models are part of the DWD reference ensemble (www.dwd.de/ref-ensemble, accessed on 20 October 2022).

### 2.3 Soil erosion modelling of the Elbe River basin

To study the effects of changing rainfall erosivity on soil erosion, we used the Water and Tillage Erosion Model and Sediment Delivery Model (WaTEM/SEDEM, Van Oost et al., 2000; Van Rompaey et al., 2001; Verstraeten et al., 2002) that was already
applied to the Elbe River basin by the authors of this study (Uber et al., 2022). The Elbe River is one of the largest rivers in Central Europe and its basin has a size of 148,300 km$^2$ located mainly in Germany and the Czech Republic (ICPER, 2016). Erosion rates are highest in the central part of the catchment on agricultural areas in rolling terrain (Pohlert, 2015; Uber et al., 2022). Locally, simulated erosion rates can be as high as 0.2 mm a$^{-1}$ (Uber et al., 2022). The average annual suspended sediment load at the most downstream monitoring station at Hitzacker is approximately 600 kt a$^{-1}$. Contaminated legacy sediments from
industry and mining as well as present-day sediment contamination from diffuse and point sources are major threats for the water quality of the Elbe River (ICPER, 2014). Furthermore, in the Lower Elbe, the estuary and the German Bight, eutrophication and oxygen depletion impair the water quality (Bergemann et al., 1996; Schöl et al., 2014; OSPAR Commission, 2017; Ritz and Fischer, 2019). Thus, the goal of a good ecologic status set by the European Water Framework Directive is not accomplished (ICPER, 2014). A further problem is the deposition of fine sediments in zones of low flow where dredging is
necessary. The waste management of the usually contaminated dredged material is very expensive (ICPER, 2014).

WaTEM/SEDEM calculates average annual soil erosion E [kg m$^{-2}$ a$^{-1}$] based on the USLE:

$$E = R \times K \times LS \times C \times P \tag{4}$$

where R is the rainfall erosivity factor [MJ mm m$^{-2}$ h$^{-1}$ a$^{-1}$], K is the soil erodibility factor [kg h MJ$^{-1}$ mm$^{-1}$], LS is the slope length factor [-], C is the crop factor [-] and P is the erosion control practice factor [-]. Downslope sediment transport is
simulated with a transport capacity (TC) approach which calculates TC [kg m$^{-1}$ a$^{-1}$] as:





$$TC = k_{TC} \times R \times K \, (LS - 4.08s^{0.8})\tag{5}$$

where $k_{TC}$ is a transport capacity coefficient [m] and s is the slope gradient [m m$^{-1}$]. $k_{TC}$ has to be calibrated. For each grid cell, it is determined whether TC exceeds the erosion rate plus possible input from neighboring cells. If this is the case, all eroded particles get transported downstream. If this is not the case, the difference gets deposited at the respective cell.

The soil erodibility factor K was taken from Panagos et al. (2014). The slope length factor was calculated with the Algorithm by McCool et al. (1989) from the digital elevation model EU-DEM of the European Environment Agency (Copernicus Land Monitoring Service, 2016), which was aggregated to a resolution of 100 m. The crop factor was obtained via reclassification of the Corine Land Cover data (Copernicus Land Monitoring Service, 2019) using the crop factor values for different land use classes given by Panagos et al. (2015a). Because no information on erosion control practices is available, the P factor was set

to one in the entire catchment. In this study, we aim at identifying the impact of climate change; thus, only changes in the R-factor are considered while changes in futue land use are not considered here.

The model was calibrated and validated with average annual suspended sediment loads estimated at 193 measurement sites in the Elbe River basin using a multi-objective optimization technique and stochastic modelling. For further details on the spatial input data, the calibration and the validation of the model, see Uber et al. (2022).

**3 Results and discussions**

**3.1 Past, present and future rainfall erosivity**

**3.1.1 Rainfall erosivity maps**

The average annual rainfall erosivity maps for the five data sets show a consistent spatial pattern (Fig. 2), which is mainly driven by topography. In all datasets erosivity is lowest in the lowlands of the North European Plain and highest in the Alpes.

In the past and present, erosivity in the lowlands is on average about 50 – 90 N h$^{-1}$ a$^{-1}$. In the Alpes, it ranges on average between 260 and 290 N h$^{-1}$ a$^{-1}$ and in the lower mountain ranges it is on average about 90 – 130 N h$^{-1}$ a$^{-1}$. The mean of the entire modeling domain is about 90 – 96 N h$^{-1}$ a$^{-1}$. This value increases to 119 N h$^{-1}$ a$^{-1}$ in the near future and 150 N h$^{-1}$ a$^{-1}$ in the far future. These maps are available on Zenodo (Uber et al., 2023) and can be used as R-factor maps in USLE based soil erosion modelling.





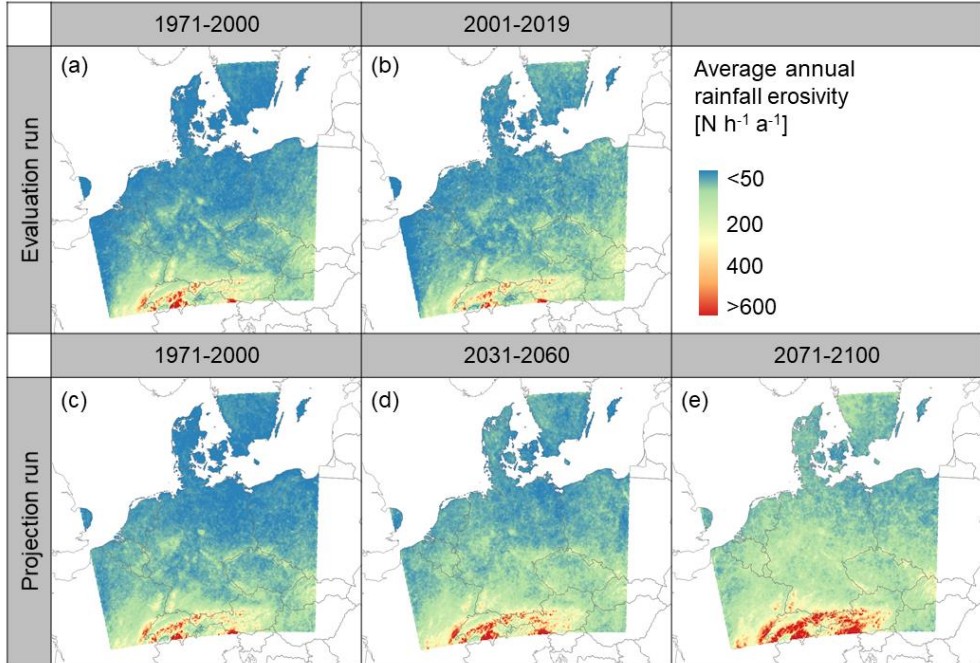

**Figure 2: Average annual rainfall erosivity (R-factor) in Central Europe in the past, present and future derived from the evaluation run (a-b) and the historic and future projection simulations (c-e).**

The maps for the past calculated from the evaluation run and the projection run are very similar (Fig. 2a and c). The spatial mean of difference between the maps is 4.9 N $h^{-1}$ $a^{-1}$ and the values for all grid cells extracted from the two maps correlate well ($R^2 = 0.91$, Fig. S1 in the supplementary material).

Beyond erosion modelling, rainfall erosivity can also be regarded as an index of heavy rain which combines rainfall intensity and cumulative precipitation depth. As such, it can also provide valuable information for other hydrological applications dealing with extreme rainfalls such as the assessment of (future) risks for flash floods or landslides or identifying zones that are prone to these natural risks (Fiener et al., 2013; Panagos et al., 2015c).

**3.1.2 Comparison to other rainfall erosivity maps**

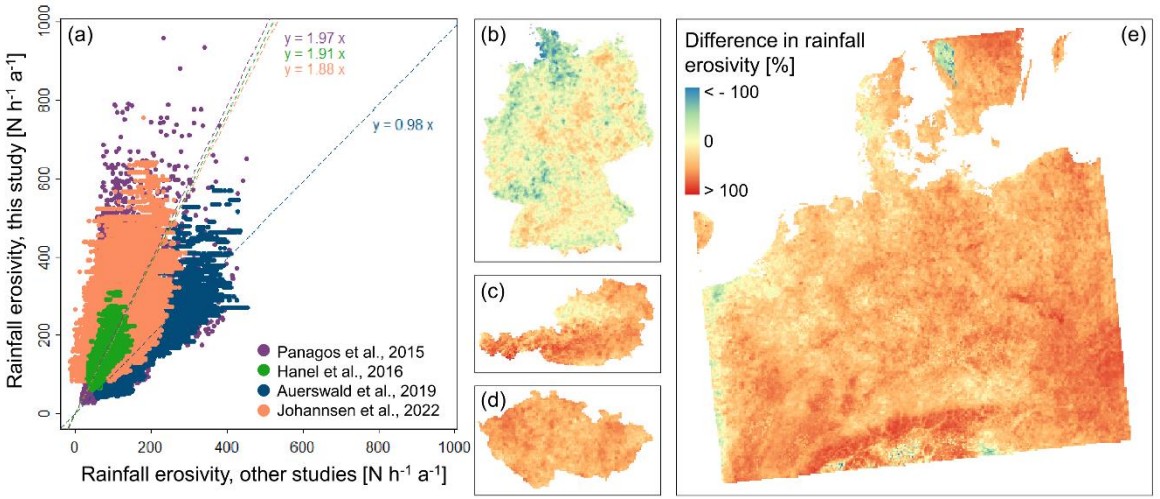

**Figure 3: Comparison of the present rainfall erosivity map generated here (evaluation run, data from 2001-2019) with maps presented by other authors. (a) Each point corresponds to a raster cell. The dashed lines show the linear models fit to the data. (b-e) show maps of differences of the map presented here and (b) the map for Germany by Auerswald et al. (2019a) covering the years**
**2001-2017, (c) the map for Austria covering the years 1995-2015 by Johannsen et al. (2022), (d) for the Czech Republic from 1989-2003 by Hanel et al. (2016b) and (e) for Central Europe covering 1970-2017 with a predominance of the last decade by Panagos et al. (2015c).**

Past and present rainfall erosivity can be compared to other available rainfall erosivity maps. Figure 3 shows that rainfall erosivity calculated from the evaluation simulation for 2001-2019 agrees well with the rainfall erosivity map by Auerswald et

al. (2019a). The correlation between the values at the raster cells is very good ($R^2 = 0.94$) and the slope of the linear regression model is 0.98, i.e. very close to one. Thus, there is no systematic difference between the two data sets and the spatial structure corresponds well to the one found by Auerswald et al. (2019a). There are regional differences nonetheless. Rainfall erosivity in the very north of Germany and in the northwest are underestimated here when compared to Auerswald et al. (2019a) and overestimated in parts of eastern Germany, the Black Forest and in the Alps (Fig. 3b). The highest values reported here (>500

N h$^{-1}$ a$^{-1}$) are not found by Auerswald et al. (2019a). This might be due to the overestimation of extreme precipitation in COSMO-CLM (Rybka et al., 2022, see Sect. 3.4). Compared to the other rainfall erosivity maps for Europe (Panagos et al., 2015c), the Czech Republic (Hanel et al., 2016b) and Austria (Johannsen et al., 2022), our values are on average about two times higher than the ones of the other authors. Nonetheless, the correlation is good ($0.85 - 0.96$), so the spatial patterns agree well. In general, differences are highest in the mountains and lower in the plains. Here, we did not correct the precipitation

data for snow (i.e. not considering precipitation during days below 0°C) as Johannsen et al. (2022) and Hanel et al. (2016b) did. This can explain parts of the differences, especially in the mountains. The differences can also be due to different temporal coverage of the precipitation data used for the generation of the maps. The temporal coverage of our map (2001-2019) is very similar to the one of the map by Auerswald et al. (2019a) (2001-2017), but agrees less with the temporal coverage of the maps of the other authors (1995-2015 for Johannsen et al. (2022), 1989-2003 for Hanel et al. (2016b) and 1970-2017 with a





predominance of the last decade for Panagos et al. (2015c)). Differences in temporal coverage are especially important in the
presence of trends. Furthermore, our methodology is very similar to the one of Auerswald et al. (2019a) (calculation from
contiguous data, hourly precipitation data, same temporal scaling factor, same equation used to calculate $e_{kin,i}$) while it differs
from the methodology used by the other authors.

### 3.1.3 Seasonal distribution of rainfall erosivity

The seasonal distribution of rainfall erosivity shows a clear peak in the summer months (late May – August, Fig. 4) and minima
from November to March. This seasonal pattern is coherent with the results obtained by Johannsen et al. (2022) for Austria,
by Auerswald et al. (2019a) for Germany and by Meusburger et al. (2012) for Switzerland. There is a strong variability from
one day to another and between subregions of the modelling domain (light grey lines and dashed black line in Fig. 4). This is
coherent with the observations made by Auerswald et al. (2019a) and can be explained by the effect of extreme rains that occur
during the same day on several pixels (Auerswald et al., 2019a). Thus, single extreme rainfalls influence the mean values
despite the large number of pixels and the long averaging period of 30 years.

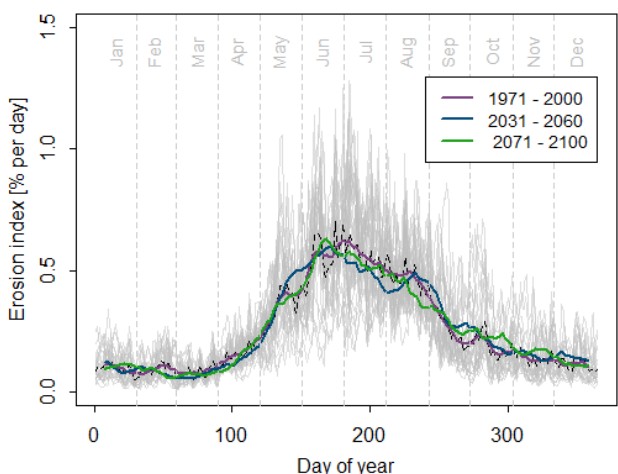

**Figure 4: Seasonal distribution of the erosion index. The light grey lines show daily erosion indexes averaged over 30 years in the past (CPS-hist, 1971-2000) and in 25 subregions of the modelling domain. The dashed black line is the average of the entire modeling**
**domain in the past and the colored lines show the 13-day moving average for each one of the data sets for the past (CPS-hist, 1971-2000), the near future (CPS-scen-nf, 2031-2060) and the far future (CPS-scen-ff, 2071-2100).**

The smoothed distribution of the erosion index does not differ considerably between the past, the near future and the far future
(Fig. 4). A comparison between past and present rainfall erosivity in Germany by Auerswald et al. (2019b) showed however,
that winter erosivity increased considerably.  In Switzerland on the other hand, Meusburger et al. (2012) observed a decreasing
trend in rainfall erosivity in February despite an increase from May to October. Thus, there might be regions in which seasonal
shifts occur, but that average out over the larger modelling domain used in this study.





## 3.2 Past and future changes in rainfall erosivity

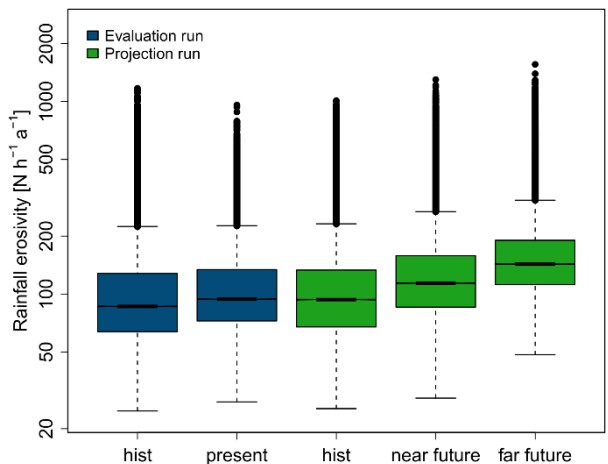

**Figure 5: Distribution of average annual rainfall erosivity (R-factor) [N h$^{-1}$ a$^{-1}$] in the five data sets.**

In the evaluation run, average annual rainfall erosivity increased between the past (1971-2000, mean: 90.5 N h$^{-1}$ a$^{-1}$) and the present (2001-2019, mean: 97.8 N h$^{-1}$ a$^{-1}$) (Fig. 5). In the projection runs driven by the global climate model, rainfall erosivity increased considerably. This is the case for all statistics (Fig. 5). Relative changes in average annual rainfall erosivity (in %) between the historical period and the near or far future are highest in the central and northern part of the modeling domain, i.e. in the river basins of the Weser, Ems, Elbe and the coastal basins in the north (Fig. 6a and 6b) where rainfall erosivity in the

far future can be up to 84 % higher than in the past. Absolute changes on the other hand are highest in the basins of the Rhine (28 N h$^{-1}$ a$^{-1}$ in the near future and 78 N h$^{-1}$ a$^{-1}$ in the far future) and the Upper Danube (37 and 74 N h$^{-1}$ a$^{-1}$ respectively). These are very strong changes that lead to a strong increase in future soil erosion rates (Sect. 3.3). Furthermore, the changes in rainfall erosivity calculated from convection-permitting climate model output are considerably higher than the ones calculated with the low-resolution approach using mean annual precipitation from model output of conventional regional climate model

ensembles (Fig. 6c and 6d).



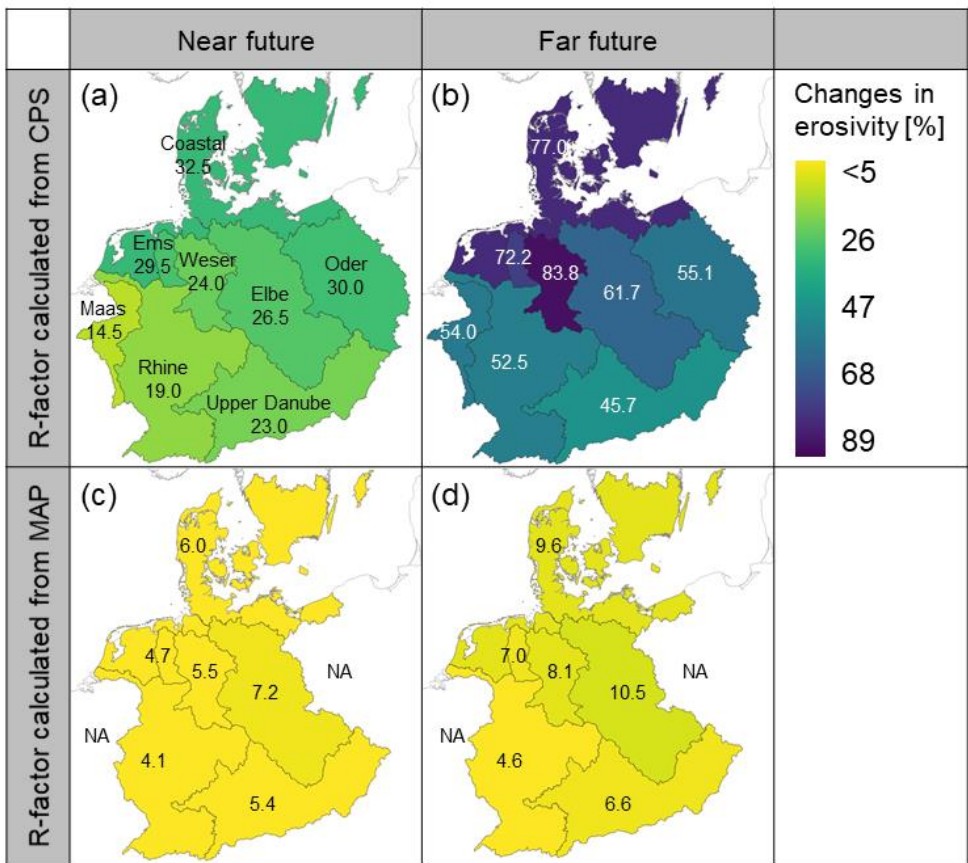

**Figure 6: Relative changes in average annual rainfall erosivity (R-factor) in the major Central European river basins between the historical period (1971-2000) and the near future (2031-2060, (a) and (c)) or the far future (2071-2100, (b) and (d)). All values are given in percent of the erosivity of the historical period. The top row shows changes in erosivity calculated with the convection-permitting simulations (CPS); the bottom row shows changes in erosivity calculated with mean annual precipitation (MAP) obtained from the median of 21 regional climate models. All simulations used emission scenario RCP 8.5.**

This finding is in line with the results of Gericke et al. (2019) who also note that the low-resolution approach underestimates rainfall erosivity and future changes in erosivity. The regression equation of the German DIN 19708 that was used here (Eq. 3) was established in the early 1990s with climate data from the 1960s to 1980s. Thus, its age and the fact that it does not consider heavy precipitation raise concerns that the equation can be transferred to the future (Gericke et al., 2019). It has to be noted that DIN 19708 explicitly states that whenever possible, high frequency precipitation should be used and that using Eq. (3) should only be used when only monthly or annual precipitation is available.

Annual rainfall erosivity in all topographic regions of Central Europe (coasts, plains, low mountain ranges, Prealps and Alpes) shows a strong interannual variability and clear trends (Fig. 7). The high interannual variability observed here is consistent with the findings of other authors who observed strong interannual variability in rainfall erosivity calculated from measured precipitation data (e.g. Verstraeten et al., 2006; Meusburger et al., 2012; Fiener et al., 2013). The presence of trends supports





the conclusions made by other authors that rainfall erosivity maps have to be frequently updated because old rainfall erosivity maps no longer represent current precipitation characteristics (Auerswald et al., 2019b; Johannsen et al., 2022).

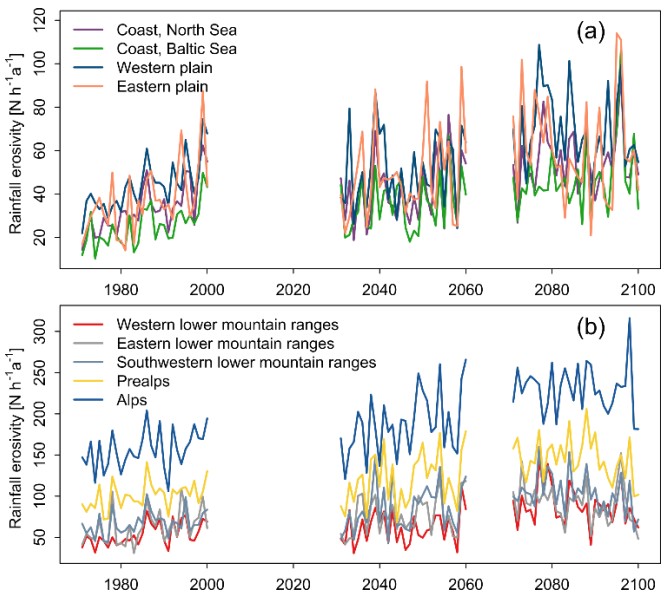

**Figure 7: Trends and interannual variability in rainfall erosivity in the natural regions of Central Europe (coast and plains (a) as well as lower mountain ranges, Prealps and Alps (b)). Rainfall erosivity was calculated with precipitation data from the projection run. The natural regions were defined according to Bundesamt für Naturschutz (2017) for Germany and were manually extended to include Central Europe based on elevation here. They are outlined in Fig. 1.**

While calculating future rainfall erosivity from CPS offers the advantage of the direct calculation from high-resolution data, it

only represents future projections from one model and one emission scenario which is less robust than using model ensembles. Thus, we compared the past and future changes calculated here to observed and simulated trends in rainfall erosivity in central Europe reported in the literature (Table S1, supplementary material). Both in the past and in the future, the range of reported trends is very large. The values given here agree well with reported values in some cases (e.g. approx. 20 % increase per decade in the Ruhr area in Germany calculated here for the projection run and reported by Fiener et al. (2013) in the period 1973-

2007). In other cases, they are strongly over- or underestimated (Table S1). It also has to be noted that for the period 1971-2000 where we estimated rainfall erosivity from the projection run as well as from the evaluation run, the trends in the two data sets can differ considerably. Usually, the changes were stronger in the projection run than in the evaluation run.

The high range of trends reported in the literature shows the need to consider model ensembles and to conduct sensitivity analyses to differences in methodology in future research. Even though the changes calculated here from CPS are considerably

higher than the ones estimated previously with a low-resolution approach and conventional regional climate model output, a comparison with the literature suggests that actual future changes could even be higher than reported here. Such strong changes in rainfall erosivity in the order of > 10 % per decade would have important implication for future soil erosion as well as for the occurrence of other natural risks such as landslides and flash floods that are triggered by heavy rain events.





### 3.3 Simulated soil erosion rates in the Elbe River basin

In the Elbe River basin, soil erosion rates are highest on rolling arable land in the central part of the catchment as well as in the Czech part of the basin. This spatial pattern is found in both sets of simulations (R-factor calculated with data from mean annual precipitation (MAP) or CPS) (Fig. 8). The order of magnitude of erosion rates is also the same, because both realizations of the soil erosion model are calibrated with the same data set of measured sediment loads. However, as rainfall erosivity calculated from CPS is considerably higher than the one calculated from MAP, the optimized values for the transport capacity

coefficient are approx. two times higher when rainfall erosivity was calculated from MAP compared to calculations from CPS.

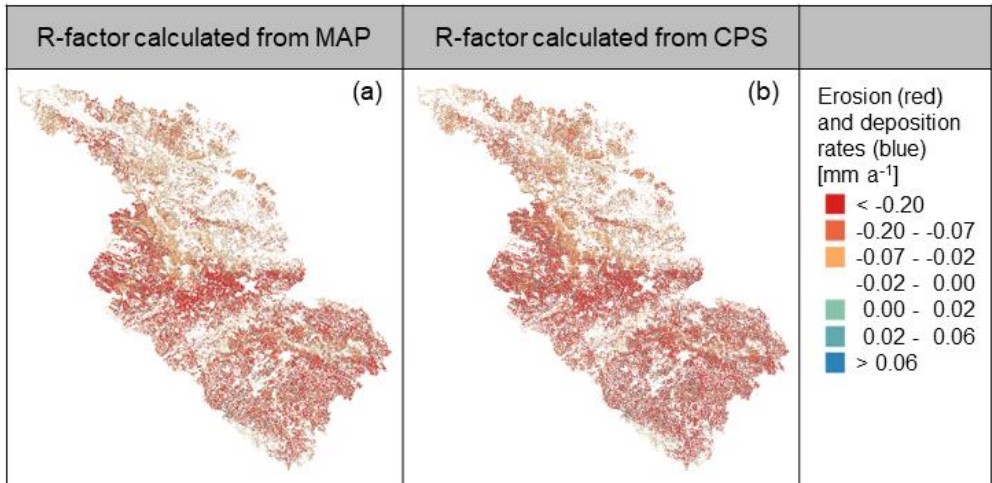

**Figure 8: Past (1971-2000) erosion and deposition rates in the Elbe River basin simulated with WaTEM/SEDEM and R-factors calculated from mean annual precipitation (a) or convection-permitting simulations (b).**

The future projections of soil erosion rates, on the other hand, differ a lot between the two model realizations. When future

erosivity is calculated from MAP obtained from the median of 21 climate models in the DWD reference ensemble run with the emission scenario RCP 8.5, erosion rates increase by 4.4 – 9.2 % in the subbasins of the Elbe River basin in the near future and by 7.4 – 12.3 % in the far future (Fig. 9c and d). These values are considerably higher when future erosivity is calculated from CPS (10.5 – 48.5 % in the near future and 34.2 – 119.5 % in the far future, Fig. 9a and b). The changes (in percent of historic erosion rates) are highest in the northern part of the basin where erosion rates are low due to the flat terrain. But also

in the subbasins of the Saale, Mulde and German Upper Elbe where erosion rates are high, erosion rates are projected to change by 19.3 – 29.1 % in the near future and 62.2 - 69.0 % in the far future. In these subbasins, absolute changes are highest.





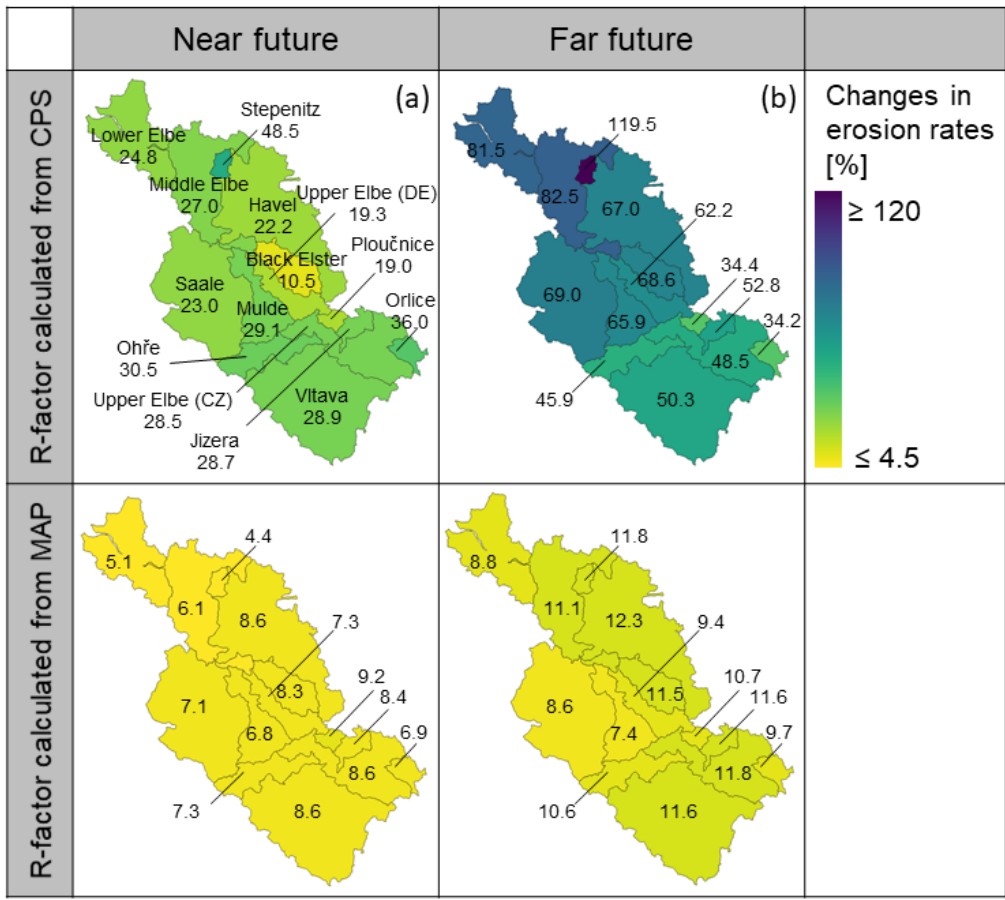

**Figure 9: Simulated relative changes in soil erosion rates in the subbasins of the Elbe River basin between the historical period (1971-2000) and the near future (2031-2060, (a) and (c)) or the far future (2071-2100, (b) and (d)). Rainfall erosivity is either calculated from convection-permitting simulations (CPS, (a) and (b)) or from mean annual precipitation (MAP, (c) and (d)).**

The changes in erosion rates lead to changes in sediment delivery to water bodies that are similar to or slightly higher than the change rates shown in Fig. 9. For example, at the outlet of the Vltava, sediment loads increase on average by 35.5 % in the near future and by 57.5 % in the far future and at the outlet of the Saale by 21.4 % and 70.7 % respectively. At the most downstream measurement station of the Elbe at Hitzacker, sediment loads increase by 22.9 % and 69.0 % respectively. It has to be noted that the reported changes are the result of changes in rainfall erosivity alone while changes in sediment connectivity, land cover and management are not considered here.

Our results show that USLE-based simulations of future changes in soil erosion are highly sensitive to the calculation of future R-factors. A similar observation was made by Eekhout and De Vente (2020) who assessed how soil erosion conceptualizations affect soil loss projections under climate change. They compared a RUSLE-based model forced by monthly precipitation, a model based on the Modified Universal Soil Loss Equation (MUSLE, Williams and Berndt, 1977; Williams, 1995) forced by runoff and the MMF model (Morgan et al., 1984; Morgan and Duzant, 2008) forced by both precipitation and runoff. They





also found that the RUSLE-based model forced with monthly precipitation cannot represent changes in extreme precipitation. Thus, in their Mediterranean study site, the RUSLE-based model projected a decrease in soil loss while the other two models projected an increase. Here, we argue that this underestimation of future soil loss with respect to the other models is not inherent in the RUSLE but caused by the low-resolution approach of calculating rainfall erosivity. Nonetheless, it is very important to compare different model conceptualizations and to include runoff as a driver of soil erosion. Here, the use of the high spatio-temporal resolution model output of convection-permitting climate models also has a high potential for future research using coupled hydrological and soil erosion models that can include runoff as a driver of soil erosion.

**3.4 Potential and limitations of convection-permitting climate simulations for the calculation of rainfall erosivity**

The maps presented here offer a high potential for erosion modelling and climate impact studies. Due to the high resolution of 3 km, they can represent the high spatial variability of rainfall erosivity in a large domain in Central Europe. Unlike most other R-factor maps (Meusburger et al., 2012; Panagos et al., 2015c; Hanel et al., 2016b), our maps do not rely on spatial interpolation and correlation with other spatial covariates such as elevation, latitude, longitude or climate indices.

Because of the high temporal resolution of the underlying precipitation data, we did not have to rely on correlations between R-factors calculated at a high resolution and low-resolution rainfall totals such as MAP, either. Many studies find a good correlation between MAP and R-factors, suggesting that MAP is a good covariate to estimate R at locations where no high-resolution precipitation data is available. However, using empirical relations between past MAP and past R-factors to derive future R-factors is problematic because it is unlikely that these relations remain stationary in the future (Quine and Van Oost, 2020). These relations are strongly conditioned by the frequency and magnitude of rainfall events that will very likely change in a warmer climate. In several regions in Europe such as the Mediterranean (Tramblay, 2012; Blanchet, 2018) and the Carpathian Basin (Bartholy and Pongrácz, 2007), MAP is decreasing while extreme precipitation is increasing. This leads to an underestimation of future R-factors that are derived from MAP alone. In Central Europe, both MAP and extreme precipitation are expected to increase (Jacob et al., 2014; Brienen et al., 2020), so future R-factors derived from MAP are also underestimated but less severely than in the above-mentioned regions. Because changes in MAP as well as in extreme precipitation are well represented in convection-permitting simulations, they offer a valuable data source for the calculation of future rainfall erosivity.

Even when the same temporal resolution of simulated precipitation data is used, Chapman et al. (2021) find that rainfall erosivity was considerably higher and observed storm characteristics agreed better with simulated ones when a convection-permitting model was used instead of a conventional convection-parameterized one.

On the other hand, the maps presented here also have limitations. Here, we calculated rainfall erosivity from precipitation data and did not consider whether precipitation falls as rain, snow or hail, so the high erosivity of hail is underestimated while erosivity in zones where considerable amounts of precipitation fall as snow (i.e. mainly the Alps) is overestimated. As rainfall erosivity in Central Europe is highest in the summer month, we assume that the impact of snow is small and can be neglected. For the Alps this is not the case, thus the very high values calculated in this region are too high. A further limitation inherent





in the USLE is that erosivity is calculated from rainfall data alone even though runoff rate is the driving force of soil detachment and transport once rills are activated (Nearing et al., 2017). Runoff-driven entrainment of soil particles is considered in the MUSLE and other models such as WEPP (Laflen, 1991), LISEM (De Roo et al., 1996), MMF (Morgan and Duzant, 2008), PESERA (Kirkby, 2008) or the ones presented by Cea et al. (2016), Nord and Esteves (2005) and Favis-Mortlock et al. (2000), but these require more complex hydrological and hydraulic modelling, which limits their applicability in the context of larger river systems.

As our maps are calculated from model output and not from precipitation measurements, the uncertainties of the model are propagated to the rainfall erosivity maps. The precipitation data was quality controlled and compared to radar- and station-based precipitation data from the past but not bias-corrected. The data showed a good agreement for extreme rainfall intensities for durations of more than 12 h but an overestimation for hourly extreme precipitation intensities (Rybka et al., 2022). Thus, it is important to compare the R-factors calculated here to the ones calculated from measured rainfall data.

Concerning the future projections, it has to be noted that current climate models struggle with estimates of future precipitation and biases are much larger than those for future temperatures (e.g. Slingo et al., 2022). Ensembles of global and regional climate models show a high range of future trends in precipitation that cannot be represented by a single model. Other studies estimated future R-factors from ensembles of global or regional climate models such as the CMIP5 model ensemble, the EURO-CORDEX ensemble or the DWD reference ensemble (Gericke et al., 2019; Panagos et al., 2022; Uber et al., 2022). In this way, the high range in projections can be represented and the uncertainty due to the choice of climate model and emission scenario can be assessed. To date, this is not possible for convection-permitting climate models because there are no model ensembles of multi-decadal simulations over large domains yet available. However, there are promising flagship studies such as the CORDEX-FPS where a first multi-model convection-permitting ensemble for the Alpes and the Mediterranean is presented (Coppola et al., 2020). Thus, the soil erosion modelling community should follow closely the coming advances in convection-permitting modelling to take advantage of new climate simulations for climate impact studies.

**4 Conclusions**

We calculated rainfall erosivity (quantified as the USLE R-factor) in Central Europe in the past (1971-2000), present (2001-2019), near future (2031-2060) and far future (2071-2100) from convection-permitting climate simulation (CPS) output. From this work, we draw three main conclusions:

- Thanks to the high spatio-temporal resolution of CPS (in this case 3 km, 1 h), R-factors can be calculated directly without having to rely on spatial interpolation and regression with aggregated precipitation sums such as mean annual precipitation (MAP). Thus, CPS offer a high potential for the calculation of future R-factors for climate impact studies on soil erosion. For the present, the R-factor map presented here is very similar to the map by Auerswald et al. (2019) that was calculated from radar-derived precipitation data.



- In the river basins in Central Europe, changes in rainfall erosivity between the past and the near future can be as high as 33 % and in the far future it can be up to 84 % higher. In the Elbe River basin, this leads to much higher rates of soil erosion (locally up to 120 % increase). These rates of change are much higher than estimated previously using regression with MAP (up to 7 % higher rainfall erosivity in the near future, 10 % in the far future). This is due to the
fact that the intensification of extreme precipitation is not represented by changes in MAP. This indicates that correlations between R-factors and MAP that were developed in the past are not necessarily valid in the future.

- A major limitation of CPS is their high computational demand. Thus, model domains are usually limited to much smaller spatial extents than the ones covered by global or regional climate models or simulated time periods are limited to short time periods. The simulations in COSMO CLM cover a long time period (in total 109 years) and a
comparably large modeling domain of approx. 1.6 million $km^2$ on land. However, to date no ensembles of CPS are available at the regional scale and for long time periods. Thus, in contrast to global or regional climate models, the uncertainty in future R-factors due to the choice of climate models cannot yet be estimated by using a bandwidth of model ensembles. Promising advances in the CPS community – including flagship studies on CPS model ensembles – suggest that in the future more CPS will be available for climate impact studies on soil erosion.

**Data Availability**

COSMO-CLM model output (e.g. hourly precipitation) is freely available from the evaluation simulations CPS-eval (Brienen et al., 2022), the historical projection simulations CPS-hist (Haller et al., 2022a) and the scenario projection simulations CPS-scen (Haller et al., 2022b) at https://esgf.dwd.de/projects/dwd-cps/ (accessed 10 February 2023). The rainfall erosivity maps presented in Fig. 2 are available at https://doi.org/10.5281/zenodo.7628957 (accessed 20 march 2023, Uber et al., 2023).

**Author contribution**

MU and GH conceived and designed the study with contributions by all coauthors. MU performed the analyses and calculations with contributions by TH, CB and MH. MH performed simulations with COSMO-CLM and provided data. MU wrote the original draft with contributions by MH. MU created the figures. GH, TH and CB reviewed and edited the draft. GH acquired funding and was responsible for project administration.

**Competing interests**

The authors declare that they have no conflict of interest.



**Acknowledgements**

This research was funded by the German Federal Ministry for Digital and Transport Network of Experts. We want to thank our colleagues at the Federal Institute of Hydrology (BfG) and DWD as well as the members of the Network of Experts

Themenfeld 1 for the fruitful discussions. R-factor calculations were run at the BfG's high-performance computers. We thank the maintainers and users for the provision of the infrastructure and useful advices. The COSMO-CLM is a regional climate model maintained by the CLM-Community. We thank the community members for their support.

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
