# Peer review of "Past, present and future rainfall erosivity in Central Europe based on convection-permitting climate simulations"

_Hydrology and Earth System Sciences, 2023_

## Referee Comment (RC3)

Reviewer comment on manuscript "Past, present and future rainfall erosivity in Central Europe based on convection-permitting climate simulations" by Uber et al.

The manuscript is very interesting and I enjoyed reading it. The authors test convection-permitting climate simulations (CPS) on their ability to predict rainfall erosivities in Central Europe. The method is of relevance for the soil erosion community and readers of HESS. I generally encourage the publication of the manuscript in HESS after minor revisions. I see some room for improvement in (i) general information about CPS and (ii) clarifying the different data products that were compared.

(i)     Many readers are potentially not familiar with the latest state of the art in climate simulations. CPS may bridge the gap between high spatio-temporal resolution that is relevant for process-based studies and long-term and large-scale studies. Hence, readers from the community of process-based studies probably appreciate information on the CPS product like: what input data is required for CPS, how are they computed, what is the methodological reason for them to perform better etc. Some of this could be added to the Introductions section L100ff, while the implementation of the CPS should be strictly placed in section 2.1.

(ii)    Within the paper a lot of different data sets and simulation periods were compared against each other. Assisting the reader with some reductions a structure might help to keep an overview.

Introduction:

The state of the art is well highlighted, but I miss more information on CPS that helps to understand their methodological advantage and why this is the case (see above).

Material and methods:

Numerous different simulation periods, (eval. and proj.) runs, rainfall kinetic energy - intensity (KE-I) relations, external erosivity products make it hard to get the main message. It might be worth considering to reduce the number of different comparisons to focus on the main message.

An overview table providing information (name, period, spatial and temporal resolution, unit, type of data set (map, table), reference, etc.) on the different data sets that were compared might help a lot.

I cannot find a definition of the evaluation and projection run. It is not fully clear to mean what was done here.

Can you provide a rough estimate on the sensitivity to use the KE-I relation by Wischmeier and Smith 1978? There are many but some are more frequently used like the exponential function by Brown and Foster 1987 as it is suggested in the RUSLE by Renard et al. 1996.

Results and discussions:

A brief discussion on the usability of CPS for landscape or field scale studies could be interesting. Just out of curiosity, how good does the CPS work for specific points in space? How good is a comparison against long term rainfall gauges?

I do not understand the point of calibrating the transport capacity to end up with same soil redistribution rates. In section 2.3 - L218 the reason to apply WaTEM/SEDEM is named as "To study the effects of changing rainfall erosivity on soil erosion [...]". From my perspective the benefit to apply WaTEM/SEDEM is to get a rough number on the differences in soil redistribution and sediment delivery to the stream network in Mg ha$^{-1}$ yr$^{-1}$. On the scale of Fig. 8, the differences between the realisations are not visible. From my perspective, Figure 9 provides a good relative number of the effect in R-factor calculation in CPS and MAE.

The drawback of using a single CPS instead of an ensemble is highlighted multiple times throughout the manuscript. Would it make sense to assess if the model tends to under or overpredict rainfall erosivity in a comparison against rainfall ground observations (rain gauge or laser-distrometer)? See comment on comparison against rain gauge point data above.

Thank you for this nice piece of work!

---

## Author Comment (AC1)

Dear Panos Panagos,

thank you very much for your interest in our study and your helpful comments on the manuscript. We agree that a considerable part of the differences between our estimates of rainfall erosivity and yours may be due to the different methodologies. This was also mentioned in the manuscript (l. 296-298). You are right that the two points that you mentioned are important and have to be discussed in more detail.

In order to quantify the effect of using a different equation to calculate specific kinetic energy from rainfall intensity, we used a subset of our data (about 8% of the model domain located partly at the coast and partly in the Alps, covering 30 years from 1971-2000) to recalculate rainfall erosivity with the RUSLE equation (Brown and Foster, 1987) that was used in Panagos et al. (2015). Thus, we could compare the results obtained with the RUSLE equation to the ones obtained in our study (using the original USLE equation (Wischmeier and Smith, 1978)) to isolate the effect of using different equations.

Rainfall erosivities in $N\ h^{-1}\ a^{-1}$ calculated with both equations are plotted in the left figure below. The USLE equation, which was used in our study, yields values that are on average 1.23 times higher than values obtained with the RUSLE equation. The right figure shows the cumulative effect of using a different equation and different temporal scaling factors. Here, our values, which were calculated with a scaling factor of 1.9, are 1.49 times higher than values calculated with the RUSLE equation and a temporal scaling factor of 1.56, as was done by Panagos et al (2015).

In total, our values are on average about 2 times higher than the values presented in Panagos et al., 2015. This difference is due to several factors (precipitation data obtained from a climate model vs. observed precipitation data, different temporal coverage of the data, differences in the methodology including the use of different equations and scaling factors). Thus, the two points that you raised explain about half of the difference between our study and the values presented in Panagos et al. (2015). We will add this important finding in a revised version of the manuscript.

Nonetheless, we wish to keep Fig. 3 in the manuscript as it is. We think that it is important to show the differences between the different available rainfall erosivity maps. We will extend the discussion on the reasons for this difference, including a discussion on the important effect of the two factors that you pointed out.

Best regards,

Magdalena Uber on behalf of all authors

References:

Brown, L. and Foster, C.: Storm erosivity using idealized intensity distributions. TRANS ASAE, 30(2), 378-386, 1987.

Panagos, P., Ballabio, C., Borrelli, P., Meusburger, K., Klik, A., Rousseva, S., Tadić, M. P., Michaelides, S., Hrabalíková, M., and Olsen, P.: Rainfall erosivity in Europe, SCI TOTAL ENVIRON, 511, 801-814, https://doi.org/10.1016/j.scitotenv.2015.01.008, 2015.

Wischmeier, W. H. and Smith, D. D.: Predicting rainfall erosion losses-a guide to conservation planning, United States Department of Agriculture, Agriculture Handbook No. 537, 58 pp., 1978.

---

## Author Comment (AC2)

Dear referee #1,

Thank you very much for your effort, your constructive feedback and your interest in our study. Your comments are very much appreciated and helped us to improve the manuscript. In the following, your comments are written in black and our responses in blue. Citations from the paper are given in italics.

Best regards,

Magdalena Uber on behalf of all co-authors.

In the submitted paper authors investigate past, present and future rainfall erosivity in relation to soil erosion in Central Europe. Authors also compare rainfall erosivity maps derived using 1h precipitation data and maps derived based on the annual precipitation data with the consideration of simple empirical DIN equation. Soil erosion-sediment transport modelling is also conducted using the WaTEM/SEDEM model. The topics is very interesting and within the scope of the HESS journal. The paper is very well written, easy to follow. I only have some moderate comments and suggestions.

Thanks for your positive evaluation of the manuscript and for your comments and suggestions.

Firstly, authors only used the RCP8.5 scenario but this is only mentioned a few times in the manuscript. Authors should definitely state this more clearly in abstract and conclusions since probably RCP2.6 and RCP4.5 would yield smaller increase in the rainfall erosivity and also in the soil erosion rates. It would be definitely very interesting to include these scenarios if input data would be available. Hence, the presented results are significantly influenced by this selection (data availability actually since COSMO-CLM (CPS-SCEN) is only available for RCP8.5). Related to this I suggest that authors add some discussion in relation to using only RCP8.5 and try to elaborate a bit more about the possible results (i.e., deviations from the presented results) using also the RCP2.6 and RCP4.5.

Indeed, it would be very interesting to compare the results obtained with RCP 8.5 to the ones that would have been obtained with RCP2.6 or RCP 4.5. Unfortunately, so far only RCP8.5 was calculated due to the very high computational demand of the convection-permitting model. There are simulations run by other groups with COSMO-CLM using other scenarios, but to our knowledge they are not yet publicly available. Furthermore, it is not sure if these results would be directly comparable to ours, because of possible differences in model versions, model configurations and the like. Thus, we can only present results with one scenario, here. We agree that results would be different if other RCP scenarios were used but we can only speculate how these different results would look like.

Thank you for pointing out that it was not mentioned clearly enough that we only used only one scenario. We added the mention of the limitation that data is only available for RCP8.5 in the abstract (line 15), discussion (line 514-515) and conclusions (line 531), now.

Secondly, part of the results is influenced by the selection of the CMIP5 model ensemble. Since CMIP6 is also available authors should at least add some discussion about the impact of using CMIP5 instead of CMIP6. This is another selection done that has probably quite significant impact on the derived results.

We agree that the choice of model selection surely influences our results. The reason why a CMIP5 model was used is that there are CMIP6 simulations available, but so far only global climate models with a coarse resolution of about 100 km. Within the framework of (EURO-) CORDEX, the global climate models are downscaled with regional climate models to a resolution of approximately 12 km but this has only been accomplished for the CMIP5 simulations, while the ones for CMIP6 are being calculated now. This is also done with ICON-CLM (which is replacing COSMO-CLM) but it will take at least 1-2 years until convection permitting simulations are available.

It was shown that CMIP6 simulations give different results than CMIP5 at the global scale. For Central Europe they are relatively consistent concerning changes in mean seasonal precipitation and extreme precipitation (Palmer et al., 2021; Ritzhaupt and Maraun, 2023). The most notable difference is that mean summer precipitation is decreasing stronger in the CMIP6 ensemble than in the CMIP5 ensemble. This is less the case for extreme summer precipitation (Ritzhaupt and Maraun, 2023).

Following your comment, we added the sentence *"Furthermore, the latest generation of CMIP6 global climate models suggests that the decrease of summer precipitation in Central Europe might be stronger than previously estimated by the CMIP5 model ensemble (Palmer et al., 2021; Ritzhaupt and Maraun, 2023) but these global models are only being downscaled by regional models now."* (line 516-519) in the discussion.

Thirdly, authors used median of the model ensemble, can you add some additional results (e.g., 25% or 75% or 10-90% quantiles) to the Supplement in order to show what is the variability among the included models.

We changed Figure 6 so that it now includes changes in rainfall erosivity with the 15th, 50th and 85th percentile of the model ensemble. Furthermore, we added the following sentences in the main text: "Furthermore, the changes in rainfall erosivity calculated from convection-permitting climate model output are considerably higher than the ones calculated with the low-resolution approach using mean annual precipitation from model output of conventional regional climate model ensembles (Fig. 6). *This is the case not only when future MAP is obtained from the median of the model ensemble but also for the entire plausible bandwidth of models. Figure 6 shows changes in rainfall erosivity estimated with the 15th and the 85th percentile of the model ensemble. Even though this approach only considers changes in MAP and not changes in rainfall intensity, it allows an estimate of model uncertainty due to the differences between the ensemble members. The results obtained with CPS are outside of the bandwidth of the model ensemble because they also represent changes in extreme precipitation in addition to changes in MAP*" (line 377-392).

Finally, the results are also significantly influenced by the data time step (1h) since conversion factor needs to be applied. I suggest that authors add more discussion about the selected temporal scaling conversion factor (i.e., 1.9) and try to elaborate about the possible impact on the derived results (i.e., rainfall erosivity and modelled soil erosion and sediment transport rates).

Thanks for the comment. This point was also made by reviewer 2 and the two community comments in the open discussion. Thus, we added an entire paragraph on this aspect (line 323-347).

Some specific comments:

-L161-162: Please add more details.

Thanks for the comment. We added an explanation why this intermediate nesting was necessary: *"This intermediate nesting was performed because it is not advised to perform direct downscaling from global models with resolutions of approximately 100 km or coarser to the very high resolution of approximately 3 km."* (line 174 - 175)

-Figure 3: Maybe add $R^2$ to the figure as well.

It was added.

-Discussion in section 3.4 is very useful.

Thank you for your feedback.

-Figure 4: Please add more details about the Erosion Index in the Material and methods section.

We added some more details in lines 223 (the unit, % $d^{-1}$) and lines 228-230: *"The erosion index varies strongly from one day to another and between grid points. Even averaged over all grid point and over 30 years, there still is a high remaining scatter, so a 13-day moving average is used for smoothing of the curves for the three data sets".*

-L461-465: From my perspective hourly resolution is actually quite problematic especially because the applied conversion factor (only one number (fixed for the whole period)), is used for different type of rainfall events (e.g., intense storms, longer duration events). In relation to this some progress should be made in future.

We understand your criticism about using a fixed conversion factor. We agree that the conversion factor is a source of uncertainty and that the assumption of a fixed value might not be valid. This was added in the discussion now (line 333-335):

*"The assumption of a constant scaling factor for the entire model domain and the entire simulated time with different types of rain and shifting intensity patterns is certainly a simplification of reality that adds uncertainty."*

For future work it might be considered to use even higher resolution data to avoid a conversion factor. COSMO-CLM model output for the variable precipitation exists at a resolution as high as 5 minutes, but it was found that this data is quite noisy. Moreover, using an even higher resolution would mean that computation times and data size would be even higher (For info: it took about a month of computation time, partly on several computers and partly on a high-performance computer cluster to calculate the rainfall erosivity maps presented here and the size of the hourly precipitation data is about 250 GB for the entire simulated time period).

-L466-471: This is only valid for the RCP8.5. It should be clearly mentioned and discussed.

Thank you for pointing this out. We now mentioned it here (line 531) and in the abstract (line 15) and discuss the limitations of using only one emission scenario in line 508-515.

References

Brown, L. and Foster, C.: Storm erosivity using idealized intensity distributions. TRANS ASAE, 30(2), 378-386, 1987.

Palmer, T. E., Booth, B. B. B., and McSweeney, C. F.: How does the CMIP6 ensemble change the picture for European climate projections? ENVIRON RES LETT, 16, 094042, https://doi.org/10.1088/1748-9326/ac1ed9, 2021.

Ritzhaupt, N. and Maraun, D.: Consistency of Seasonal Mean and Extreme Precipitation Projections Over Europe Across a Range of Climate Model Ensembles, J GEOPHYS RES-ATMOS, 128, e2022JD037845, https://doi.org/10.1029/2022JD037845, 2023.

---

## Author Comment (AC3)

Dear referee #2,

Thank you very much for your detailed review, for your interest in the study and for your valuable feedback on the manuscript. Your comments helped us to improve the paper.

In the following, your comments are written in black and our responses in blue. Changes made in the manuscript are indicated with Italics.

Best regards,

Magdalena Uber on behalf of all co-authors

The study of Uber et al. is very interesting and of high importance for proper estimation of future rainfall erosivity and future soil loss in a changing climate. The topic of the submitted manuscript addresses well the scope of the HESS journal. Uber et al. use precipitation data from convection-permitting simulations (CPS) for estimation of future rainfall erosivity in Central Europe. The use of CPS-based precipitation data for estimations of future erosivity is the novelty of their study. CPS include explicit simulation of large deep convection cells what is not the case in conventional convection-parametrized climate models but important for erosivity estimations. CPS-based precipitation data are available in a spatial resolution of around 3 km and a temporal resolution of 1 h. This spatial and temporal resolution of precipitation data is considerably higher than it is the case for the output of convection-parameterized models. A high spatial and temporal resolution of precipitation is also of importance for erosivity estimations. Uber et al. describe well in their introduction the potential of CPS-based precipitation data for improved erosivity estimations for the near and far future and appropriately explain the limits of current CPS outputs for it. The major limit of CPS-based precipitation data for erosivity projections is the lack of ensembles of CPS. This lack results in an unknown uncertainty of erosivity projections. So far comprehensible, Uber et al. try to account for this by comparing single-CPS-based erosivity estimates with, first, erosivity estimations with precipitation data from CPS evaluation simulations driven by observation data, and, second, with erosivity estimations by a regression equation based on mean annual precipitation sums using precipitation data from conventional convection-parametrized climate model ensembles. Although their attempt is much appreciated, it must be pointed out that this equation (Equation 3 in the manuscript) is explicitly not applicable to current and future erosivity estimations. This is described in the DIN 19708:2022-08. In consequence, the validation approach of comparing single-CPS-based erosivity projections with erosivity projections based on Equation 3 of the manuscript seems incorrect. This approach might be possible for the past period but should be carefully applied due to the trend of increasing erosivity that has already existed in recent decades.

Thanks for your comment. We certainly agree that the use of the DIN-equation (Equation 3) established with past data for projections of future rainfall erosivity has many flaws and should be avoided. However, due to a lack of high-resolution data, it is quite common to estimate future rainfall erosivities based on regressions with lower resolution rainfall totals that were established in the past. We did not intend to validate our results by comparing them with the results obtained with the projections based on mean annual precipitation (i.e. results generated with Equation 3). Instead, we wanted to show that the changes in rainfall erosivity projected using CPS are much higher than estimated previously with Equation 3.

We fully understand your doubts on using Equation 3 and we argue that one of the main advances of using CPS for erosion modelling is that we don't have to rely on regression

equations such as Equation 3 anymore. Nonetheless, we think that using Equation 3 also has advantages over using CPS, the main one being that it can easily be applied to (projected) climate data of basically any temporal resolution. In our case, it enables an estimate of changes in rainfall erosivity due to changes in mean annual precipitation from an ensemble of climate models. Thus, variability between climate models can be assessed which is not yet the case for CPS.

Apparently, we did not point this out clear enough in the manuscript. Thus, we changed Figure 6 so that it now includes results showing the bandwidth of the climate model ensemble. Furthermore, we added the following paragraph (line 377-392):

"Furthermore, the changes in rainfall erosivity calculated from convection-permitting climate model output are considerably higher than the ones calculated with the low-resolution approach using mean annual precipitation from model output of conventional regional climate model ensembles (Fig. 6). *This is the case not only when future MAP is obtained from the median of the model ensemble but also for the entire plausible bandwidth of models. Figure 6 shows changes in erosion rates estimated with the 15th and the 85th percentile of the model ensemble. Even though this approach only considers changes in MAP and not changes in rainfall intensity, it allows an estimate of model uncertainty due to the differences between the ensemble members. The results obtained with CPS are outside of the bandwidth of the model ensemble because they also represent changes in extreme precipitation in addition to changes in MAP.*"

Uber et al. also use their CPS-based erosivity estimates for estimation of future changes of soil loss rates by water erosion and sediment delivery. For this, they select the Elbe River basin. Their USLE-based soil loss estimates for the near and far future consider changes in the R-factor exclusively. Although Uber et al. calculated the erosion index from their erosivity projections (chapter 3.1.3), they don't consider changes in the C-factor for their future soil loss estimations. Also, potential changes in crop growth due to prolonged vegetation periods are not considered or changes of the other factors.

Indeed, we focused on the R-factor and did not consider changes in the C-factor or the other factors, because -compared to the strong changes in R-Factor- they are less affected by climate change. As suggested by you below and also by Shuiqing Yin in her comment, we removed the part on soil erosion modelling in the manuscript.

Only future changes in the R-factor are considered in their erosion modelling. In consequence, estimates of future relative changes in soil loss are equal to relative changes in erosivity. This is because the USLE is a multiplicative approach. So, the purpose of the efforts of Uber et al. for the chapter on simulated erosion rates is unclear. I recommend to either consider, at least, changes in the erosion index or to strongly shorten the chapter on future erosion rates.

As suggested we removed the part on soil erosion modelling.

The manuscript would benefit from a concentration on single-CPS-derived erosivity projections and an in-depth discussion of e. g. the applicability of scaling factors accounting for spatial and temporal resolutions of precipitation data lower than those from e.g. 1-min precipitation data from rain gauges. This would also address the comment by Panos Panagos (https://doi.org/10.5194/hess-2023-120-CC1) who criticises the applied temporal scaling factor 1.9 for being too high. Panagos et al. (2015) found a lower temporal scaling factor, but

this was used to correct only to 30-min resolution. Considering in addition their temporal scaling factor to correct from 30-min resolution to 5-min resolution (they calculated no factor for 1-min resolution), it results in a factor of even slightly higher than 1.9. In consequence, the temporal scaling factor from Fischer et al. (2018) applied by Uber et al. seems to be in accordance with the temporal scaling factors found by Panagos et al. (2015). Nevertheless, the question could be discussed whether the temporal scaling factor is still valid when rainfall intensities further increase in the near and far future. Moreover, it could be of interest to explain/discuss why no spatial scaling factor was necessary to account for the underestimation of erosivity despite the spatial resolution of around 3 km. A possible reason might be the overestimation of hourly extreme precipitation intensities mentioned in the discussion (page 19, line 444).

Thank you for your very helpful comment. We included these aspects in the revised version of the manuscript by adding a paragraph on the effect of using different equations to calculate specific kinetic energy and the effect of the scaling factor as well as on the uncertainty related to the scaling factor (line 323-347).

Although the manuscript is very well written, it could benefit from a separation of the validation of precipitation and erosivity from CPS projection runs by precipitation and erosivity from CPS evaluation runs, and the results of erosivity from CPS projection runs and their discussion in context to results from other studies.

We are not sure what exactly you mean here. As the time periods that are covered by the evaluation run (1971-2019) is not identical to the one of the projection run (1971-2000 and the two future periods), the results from other studies could only be compared to the evaluation runs for the studies reporting past, observed erosivity or to the projection runs for the studies reporting projected future erosivities.

All in all, the manuscript could benefit from revisions as suggested above and in more detail in the follwoing list below.

Detailed comments:

page 2, line 32:   The citation of Nearing et al., 2017 for the definition of rainfall erosivity gives the impression that rainfall erosivity was no topic before. I suggest referring to an appropriate earlier publication from the 1950s to 1970s, at least in addition.

The definition by Nearing is very precise and we did not find such a good definition in the early publications. So we added the fact that the concept is much older in a subphrase: "Rainfall erosivity *was first quantified in the 1950s and can be* defined as "the capability of rainfall to cause soil loss from hillslopes by water" (Nearing et al., 2017)" (line 31-32).

page 2, line 38:   Same as comment above; an earlier publication is recommended at least in addition to Wilken et al., 2018

An earlier publication (Laws and Parsons, 1943) was added (line 38).

page 2, line 40/41: The "Thus," is irritating as it gives the impression that there is a conclusion, but it is not obvious from what this conclusion is taken. It doesn't seem to be taken from the preceding sentence. Please, make it clearer.

It was changed accordingly, the "Thus," was deleted.

page 2, line 42:    It might be rather the suitability of the R-factor equations to express rainfall erosivity which depends on the temporal resolution of the precipitation data.

It was changed accordingly (line 42-43).

page 2, line 47:    "This approach" is not clear as you refer in the preceding sentence to "low-resolution approaches" which suggests that there are several approaches.

Thanks for pointing this out. It was changed to plural ("*These approaches …*", line 48).

page 2, line 49/50: Please add a reference to the expectation of changes in the frequency distribution of rainfall events.

The reference is given later in the text (line 62-65). Here we think that it would be misleading as it would suggest that the reference is referring to the entire sentence.

page 2, line 56:    Every error needs to be assessed critically. This sentence has not really information. Please clarify.

Thanks for the comment. We agree and deleted the sentence. The potential and limitations of satellite-based precipitation products is not the topic of this paper.

page 2, line 57:    Please clarify that an increase in precipitation (intensity and annual total) is likely rather for wet than for dry global regions according to your reference Sun et al., 2007. Moreover, to my understanding, changes in precipitation are the result of the changes in temperature and both, changes in temperature and precipitation are the changed climate. So, changes in temperature and precipitation are not the result of climate change but they are the changed climate itself which, simplified, results from increased concentrations of greenhouse gases in the atmosphere.

We rephrased the sentence. It now says: "*Globally, precipitation is increasing due to an increase in atmospheric water vapor in warmer air (e.g. Allan et al., 2020; Fowler et al., 2021)*" (line 59-60). Of course, there are regional patterns, but we prefer not to go into the details of global precipitation changes but focus on Central Europe.

page 2, line 59/60: Please clarify what specifically is increasing. Do you mean the number of extreme events?
It is not clear what you mean with an "intensified hydrologic regime". Please edit.

It was changed accordingly. We hope that the explanation is clearer now. It now says: "Furthermore, *warming and higher atmospheric moisture fluxes lead to an intensification of the water cycle causing an increase of the intensity and frequency of extreme precipitation, globally as well as in Central Europe (Allan et al., 2020; Brienen et al., 2020; Fowler et al., 2021). Strong increases in extreme precipitation are* due to the fact that the share of convective precipitation in total precipitation is increasing (Berg et al., 2013)." (line 62-66).

page 2, line 60/61: "This is due to the fact…" suggests that you are describing a cause-and-effect relationship, but, actually, it just clarifies the preceding sentence. I would suggest combining both sentences (line 59 – 61) into one but clear sentence.

It does describe a cause-and-effect relationship because convective precipitation often causes intense precipitation events while stratiform precipitation rather causes long-lasting but less intense rain events.

page 3, line 75:   Do you mean 11 out of the 196 studies?

Yes. It was changed accordingly (line 80).

page 3, line 95:   Do you mean with "at the time scales needed…" the length of the time series? If yes, it might be clearer to write "for the length of time series" as 'time scales' may be used rather in context to the temporal resolution of the data. If you agree, please keep it in this way throughout the manuscript.

Thanks for pointing this out. It was changed accordingly (line 105, 151).

page 4, line 108:  Please add the reference where to find the published data.

The reference was added (line 118).

page 4, line 110:  Was the temporal resolution of CPS already mentioned? If not, please add so that it is clear what you mean with high temporal resolution.

The temporal resolution of the COSMO CLM output used here is given later in the text (1 h, line 188). Unfortunately, it is not possible to add a typical temporal resolution of CPS here. This is due to the fact that the time steps of the model calculation are very short (a few seconds) while the temporal resolution of the model output depends only on the choice of the modeler at which time steps the output is actually written.

page 4, line 120:  Which size of regions to do mean? In context of the globe (mentioned in the same sentence), a region could also be Europe, but you may rather think of smaller areas? Maybe it is possible to provide the typical spatial extent in km$^2$ or to refer to a specific geographic region as an example.

It now says "i)limited spatial extent of most CPS. While regional and global convection-parameterized simulations cover the entire globe, to date *CPS are only available for limited areas in most regions of the world (e.g. Central Europe) due to constraining factors like computing power.*" (line 130-132). We hope that this makes the availability of CPS clearer. They are very different in extent and exist in most regions of the world, but don't cover the entire globe.

page 6, line 170:  It should be specified that you mean the final CPS model output as you mentioned several different models before.

It was clarified. Now it says "*The COSMO-CLM CPS model output…*" (line 188).

page 6, line 172:  What is the 'FPS-convection contribution'?

It was rephrased, now saying "The overall configuration of our simulation has been taken from *a joint contribution of the CLM-community to a CPS experimental study for Central Europe* (Coppola et al., 2020)." (line 190-192).

page 7, line 182: Please add that the point that these must be 6 hours without any precipitation.

Thank you for pointing this out. It was added accordingly (line 203).

page 7, line 186: In case of $I \geq 76.2\ mm\ h-1$, the kinetic energy should be 28.33 * $10^{-3}$; please correct this factor.

Thanks for having noted this error. It was corrected (line 207).

page 7, line 194: If 'grid cell' is meant to be equal to 'grid point' then it may be good to use the same wording throughout the manuscript for the sake of simplicity.

It was changed accordingly (line 215 and elsewhere).

page 8, line 208: It is not clear why erosivity is also calculated by Equation 3. "For comparison" is not enough as a reason. Is it meant for validation of the R-factors derived from single-CPS-based precipitation data? This equation is not valid for calculation of current and future erosivity as described in the beginning.

We understand your criticism of using equation 3, see our response on your comment in the beginning. We did not mean to validate our results with this comparison but rather wanted to show that the new results presented here are considerably higher than previous estimates obtained by using Equation 3.

page 8, line223: The value 0.2 mm/a for simulated erosion rate should be based on more information. Is it a multi-year average and of which time period - past, present, future? What means 'locally'? Is it simulated for a single field? Or averaged for e.g. a county? Please specify to what this value relates.

The part on soil erosion modelling was deleted as suggested.

page 8, line 220-230: This text section may be rather part of the introduction. Think about shifting it to page five, line 131.

The part on soil erosion modelling was deleted as suggested.

page 9, line 245: You write that the aim is to identify the impact of climate change on soil erosion and therefore you just consider changes in the R-factor. But also, the erosion index changes and the soil loss ratios by e.g. prolonged growing season (even when the management itself is not changed). Climate change might also affect soil erodibility (e. g. by changes in soil organic matter). So, you should not write that you aim to identify the impact of climate change but rather the impact of changes in the R-factor by climate change on soil erosion estimates.

The part on soil erosion modelling was deleted as suggested. To answer your question nonetheless:

We agree that climate change also affects the other factors, but the main effect is on the R-factor. As in every modelling study, we had to make assumptions and simplifications that certainly introduced errors and uncertainties. So we think that – considering all the remaining

uncertainties – the impact of climate change on soil erosion is quite well represented by its effect on the R-factor.

It would be interesting to also consider the other effects of climate change that you mentioned and to attribute changes in soil erosion to the different effects of climate change. We will keep your comment in mind for future work.

page 9, line 247f: Sediment load measurements of which years were used for calibration and validation of the model? Please add this information.

The part on soil erosion modelling was deleted as suggested. To answer your question nonetheless:

The data covers the time period between 1963 and 2020 with the majority of data being available after 1990. We are aware that the sampling dates correspond not exactly to the reference period of the model but neglecting the data outside the reference period would have caused a loss of valuable data. This was discussed by Uber et al. (2022).

page 9, line 255f: Why don't you distinguish between past 1971-2000 and present 2001-2019? Would be good to have average annual erosivity of the entire modelling domain for both periods separately.

This information is given in Figure 5 and line 369-372. Here we focus on the spatial pattern where the differences between the past and the present (or more precisely the differences between CPS-hist from the projection run, CPS-eval-hist, CPS-eval-present) are smaller than spatial differences.

Moreover, don't you mean that average annual erosivity in the lowlands is between 50 and 90 N h$^{-1}$ a$^{-1}$ rather than "erosivity in the lowlands is on average about 50 – 90 N h$^{-1}$ a$^{-1}$"? The term 'on average' lets one expect a single value.

You are right, it was chanced accordingly (line 278).

page 9, line 257:  Similar to the comment above; how to interpretate the range 90 – 96 N h$^{-1}$ a$^{-1}$ as a mean? Is this range arising from a confidence interval around the mean? "The mean of the entire modeling domain" let one expect a single value which is then also given for the near and far future. Please edit respectively.

It was clarified in the manuscript. It now says "*In the past, the mean of the entire modeling domain is 91 N h$^{-1}$ a$^{-1}$ in the evaluation run (CPS-eval-hist) and 96 N h$^{-1}$ a$^{-1}$ in the projection run (CPS-hist)."* The values for the near and far future are now given in section 3.2 (line 371-372).

page 9, line 258:  You are inviting people to use your R-factor results for USLE-based soil loss modelling. But for this, the erosion index is missing. So, you would need to provide these data as well. Moreover, I like to recommend providing most important information about the data and development in a table and pointing out that the results are based on a single climate model and not on a model ensemble using the RCP8.5 scenario.

We added the phrase "*Data of the erosion index can be provided on request to the first author.*" In the Data Availability section. We are not sure which information you would like

to see in a table. We believe that you are referring to the description of the data on Zenodo. As the name of the model is given in the description, it is clear that the results are based on a single climate model. We added the fact that the emission scenario RCP 8.5 was used in the description on Zenodo. It now says "Past, present and future rainfall erosivity in Central Europe calculated from convection-permitting climate simulations in COSMO-CLM *using emission scenario RCP 8.5.*"

page 11, line 294f: Is it possible to calculate in addition the mean annual R-Factor from the evaluation runs for the respective time periods used in the other studies and to compare these? Moreover, the differences could be discussed in context to changes of the R-factor in the last decades as discovered by other studies already.

It would be possible, but then the comparison would concern a "new" data set and not the one that was published on Zenodo and that is presented here. To make the point that differences in temporal coverage lead to different results in the context of changes of the R-factor in the last decades we edited the following sentence. It now says "Differences in temporal coverage are especially important *given the observed increases in R-factors in the last decades (e.g. Hanel et al., 2016a; et al., 2019a; Auerswald et al., 2019b)*" (line 319-321).

page 12, line 300f: What is the reason for comparing results of other studies for the R-factor with the results of the evaluation run but for the erosion index with the results of the projection run?

The calculation of the erosion index took a lot of calculation time, so it was only calculated for the data from the projection run, not the one of the evaluation run.

page 12, line 300f: How does the seasonal distribution of erosivities from evaluation runs and projection runs fit together for the two periods respectively? Please add information about this.

See our answer above.

page 12, line 316f: Should it not be relatively easy to analyse the erosion index restricted to the area of the individual countries? This should be done instead of guessing that the changes in the intra-annual distribution of erosivity level out across the modelling domain.

Indeed, this possible explanation is rather a guess than based on analyzing the data. Thus, we deleted the sentence "Thus, there might be regions in which seasonal shifts occur, but that average out over the larger modelling domain used in this study". The question why the model doesn't confirm the observed changes in seasonality is still open and we will keep it in mind for further research.

page 13, line 321f: Mention the mean annual rainfall erosivities for near and far future so that it is easier to compare the results of evaluation and projection run.

The values were added (line 371-372).

page 14, line 339: It is not the age but changes in the precipitation characteristics and the fact that it does not consider rainfall intensities but only total rainfall amount.

The sentence was revised, now saying "Thus, *changes in precipitation characteristics* and the fact that it does not consider heavy precipitation raise concerns that the equation can be transferred to the future (Gericke et al., 2019)" (line 396-397).

page 14, line 341: The DIN 19708 also explicitly states that the regression equations are not suitable for calculation of R-factors of (even) the presence, not to mention the future.

We believe that you are referring to the updated DIN 19708:2022-08. At the time when we calculated R-factors using Equation 3, only DIN 19708:2017-08 was available.

page 15, line 364f: Discard the part of the sentence which refers to the comparison with the results from Equation 3.

It was changed accordingly (line 422-423).

page 15, line 366: Please add the references which report higher future increase in erosivity in comparison with your results from CPS.

The references were added (Panagos et al. (2017, 2022), line 425).

page 16, line 370f: Please indicate the period to which your results on erosion rate estimates relate here.

The part on erosion modelling was deleted as suggested.

page 17, line 391f: It is unclear what the message of the first sentence is. Relative changes in erosion rate are equal or slightly higher than relative changes in sediment delivery to water bodies? Moreover, it is not indicated to which simulations ('CPS' or 'MAP' approach?) the mentioned changes in sediment load in the near and far future relate. Please revise.

The part on erosion modelling was deleted as suggested.

page 17, line 397: With respect, your results are not necessary to know that USLE-based simulations of future changes in soil erosion are highly sensitive to changes in the R-factor when just the R-factor is changed while the other factors are assumed to remain equal.

The part on erosion modelling was deleted as suggested.

page 18, line 410f: Please indicate that the resolution of precipitation data of 3 km is just high in comparison to most other projected precipitation data (as this is not the case in comparison to measured precipitation data by e. g. ground-based radar).

As this paragraph is on the advantages and the potential of using CPS, it would be contradictory to start it with a limitation (i.e. that the resolution of CPS is still lower than the one of radar derived precipitation products). Compared to the resolution of global (~ 100 km or more) or regional climate models (in Europe ~ 12 km) or typical densities of rain gauge networks, we think that it is justified to say that the 3 km resolution of the CPS is in the same order as the 1 km resolution of typical radar precipitation products.

page 18, line 427f: The conclusion of the results from Chapman et al. (2021) cited by you would be that projected monthly (? - what is "the same temporal resolution"?) precipitation

sums are higher from the convection-permitting model than from the conventional convection-parameterized model as changes in erosivity resulting from changes in the rainfall intensity are not necessarily reflected in changes of e.g. monthly precipitation sums. What is the reason for that?

We added the resolution (3h, line 486).

page 19, line 435f: Here, you discuss limitations of the USLE although the intension of chapter 3.4 is to discuss potentials and limitations of CPS for calculation of erosivity. Delete the part on limitations of the USLE and concentrate on discussing the limitations by CPS-derived erosivities as calculated in your study. For example:

Agreed, we deleted the sentences on the limitations of the USLE.

Do you expect an underestimation of erosivity in near and far future by using a constant temporal scaling factors of 1.9 for accounting of erosivity underestimation by using 60-min data?

We added this thought in section 3.1.2 where it now says "*The assumption of a constant scaling factor for the entire model domain and the entire simulated time with different types of rain and shifting intensity patterns is certainly a simplification of reality that adds uncertainty*" (line 333-335)

and

"*[…] and the temporal scaling factor might have to be adapted to future data with higher intensities of extreme events*" (line 339-340).

In the introduction you mentioned that CPS can simulate large deep convection cells but not smaller shallow convection (line 84). Which consequence do you expect from this for projected near and far future erosivity estimates?

CPS do simulate shallow convection but it is not calculated explicitly but using a parameterization. Because shallow convection doesn't generate heavy rain, it is assumed that the parameterization doesn't have a strong effect on rainfall erosivity.

Are there any indicators which allow a guess whether your single-CPS-based results of erosivity in the near and far future are laying in the lower, middle, or upper range of erosivities derived from future ensembles of CPS?

We assume that our estimates of rainfall erosivity would be rather at the upper end of future ensembles of CPS. The reason is that the driving simulation MIROC CCLM is rather at the "wetter" end of the bandwidth of the EURO-CORDEX ensemble. Also, the overestimation of heavy rain in COSMO CLM causes high values of rainfall erosivity.

But the research on CPS is very active and a lot of effort is made in the community to optimize these models. Thus, it is still unclear how future ensembles of CPS will look like and we don't want to speculate on that without having actual data to compare our results to.

page 19, line 442f: The comparison of modelled and measured precipitation data is an important result. What is the reason that you don't provide this information earlier, e. g. when you compare your erosivity estimations from projection runs with those from evaluation runs? What is the reason for the overestimation of hourly extreme precipitation intensities? How large is this overestimation?

The comparison of modelled and measured precipitation has been presented elsewhere (Rybka et al., 2022). The overestimation of hourly extreme precipitation is probably due to the configuration of the CPS.

We agree that this has to be kept in mind when the rainfall erosivity data presented here is used. Thus, we added a sentence in line 503-504: "*This* [the overestimation of hourly extreme precipitation] *leads to an overestimation of the rainfall erosivity presented here that has to be kept in mind.* Thus, it is important to compare the R-factors calculated here to the ones calculated from measured rainfall data."

page 19, line 446: What do you mean with "estimates of future precipitation"? Do you mean hourly precipitation sums, or monthly, or yearly?

This sentence is meant as a general statement, saying that future precipitation is harder to project than future temperature.

page 20, line 468: Which spatial scale do you mean with "locally"? Is it a single 'pixel' of a certain size? How can the increase in soil erosion be higher than the increase in erosivity although all other USLE factors remain constant? Do you refer once (line 467) to an average across an area and once (line 468) to a single 'pixel'?

It referred to the subbasins shown in Figure 9. But this part was deleted in the manuscript.

According to language/grammar:

Please write either "modelling" or "modeling" throughout the manuscript.

Thank you for pointing out this inconsistency. It was harmonized.

page 2, line 34: A comma might be necessary before "and" in "…its derivates and models…".

We did not use the Oxford comma throughout the Manuscript.

page 4, line 108: Please use either singular or plural of 'data' consequently throughout the manuscript.

It was harmonized.

page 5, line 131: "…the new rainfall erosivity maps" sounds as it will be official maps. I suggest to revise this sentence. Maybe like "Furthermore, modelled rainfall erosivities of these periods were used in the USLE-based model WaTEM/SEDEM to estimate changes in soil erosion and sediment delivery to the Elbe River."

We replaced "maps" with with "data".

page 5, line 137: With 'or results' you may mean 'our results'. Please correct.

It was corrected.

page 7, line 182: Instead of "We use", it may be "We used".

It was corrected.

page 9, line 246: 'r' is missing in 'future'.

It was corrected.

page 9, line 254:  Should be 'Alps' instead of 'Alpes'. Please adjust also Fig.1 accordingly.

It was corrected.

page 9, line 255:  Should be 'Alps' instead of 'Alpes'.

Thank you for noting this error. It was corrected throughout the manuscript.

page 12, line 315: Why do you write "…despite…" here? Isn't it logical that the erosion index needs to decrease in other months when there is an increase in May to October?

You are right, we replaced "despite" with "and".

Thanks again for your thorough review. We appreciate the time and effort you took to comment on the paper. We hope that we addressed all your comments satisfactory and we are convinced that your remarks and propositions helped us to improve the manuscript.

References

DIN-Normenausschuss Wasserwesen: DIN 19708:2022-08 Bodenbeschaffenheit - Ermittlung der Erosionsgefährdung von Böden durch Wasser mit Hilfe der ABAG, https://dx.doi.org/10.31030/3365455, 2022.

Fischer, F. K., Winterrath, T., and Auerswald, K.: Temporal-and spatial-scale and positional effects on rain erosivity derived from point-scale and contiguous rain data, HYDROL EARTH SYST SC, 22, 6505-6518, https://doi.org/10.5194/hess-22-6505-2018, 2018.

Panagos, P., Ballabio, C., Borrelli, P., Meusburger, K., Klik, A., Rousseva, S., Tadić, M. P., Michaelides, S., Hrabalíková, M., and Olsen, P.: Rainfall erosivity in Europe, SCI TOTAL ENVIRON, 511, 801-814, https://doi.org/10.1016/j.scitotenv.2015.01.008, 2015.

---

## Author Comment (AC4)

Dear referee #3,

Thank you very much for your feedback, your interest in the study and your recommendation for publication after minor revisions. In the following, your comments are written in black and our responses in blue. Changes made in the manuscript are indicated with Italics.

Best regards,

Magdalena Uber on behalf of all co-authors

The manuscript is very interesting and I enjoyed reading it. The authors test convection-permitting climate simulations (CPS) on their ability to predict rainfall erosivities in Central Europe. The method is of relevance for the soil erosion community and readers of HESS. I generally encourage the publication of the manuscript in HESS after minor revisions. I see some room for improvement in (i) general information about CPS and (ii) clarifying the different data products that were compared.

(i)     Many readers are potentially not familiar with the latest state of the art in climate simulations. CPS may bridge the gap between high spatio-temporal resolution that is relevant for process-based studies and long-term and large-scale studies. Hence, readers from the community of process-based studies probably appreciate information on the CPS product like: what input data is required for CPS, how are they computed, what is the methodological reason for them to perform better etc. Some of this could be added to the Introductions section L100ff, while the implementation of the CPS should be strictly placed in section 2.1.

Thank you for pointing this out. We added some information on CPS that will be helpful for readers that are less familiar with climate modeling. It now says: "*CPS are performed with regional climate models (RCM) on a high spatial resolution (usually ≤ 4 km). Due to the coarse resolution of conventional climate simulations, deep convection has to be parameterized as a sub-grid scale process, which leads to deficits in the realistic simulation of precipitation. This parameterization is* switched off in the model *setup of a CPS (Lucas-Picher et al., 2021),* allowing the model to simulate the precipitation explicitly in each grid cell. *A good representation of deep convection is crucial because it is the main source of precipitation in many parts of the world and especially important as it often generates extreme precipitation (Prein et al., 2015).*" (line 88-95).

We think that the details on input data to the climate models and the way they are computed is beyond the scope of this paper. For the model used here, these details can be found in the references that are given in sect. 2.1.

(ii)
        Within the paper a lot of different data sets and simulation periods were compared against each other. Assisting the reader with some reductions a structure might help to keep an overview.

Thank you for pointing this out. We added a table, giving an overview of the different data sets as you proposed. We hope that this will help the reader to better understand the data sets.

Introduction:

The state of the art is well highlighted, but I miss more information on CPS that helps to understand their methodological advantage and why this is the case (see above).

We hope that the additional information in lines 88-95 (see above) will help to clarify this issue.

Material and methods:

Numerous different simulation periods, (eval. and proj.) runs, rainfall kinetic energy - intensity (KE-I) relations, external erosivity products make it hard to get the main message. It might be worth considering to reduce the number of different comparisons to focus on the main message.

Unfortunately, it is not possible to reduce the number of comparisons. It is not possible to use only the evaluation or projection runs because the latter do not provide data for the present (which is important for the comparison with observed data) while the former does not provide data for the future.
External erosivity products are also important. Because the presented data is very novel and there are considerable remaining uncertainties, it is necessary to compare it to more established data sources.
Besides for the response to Panos Panagos' comment, we did not compare different KE-I relations.

An overview table providing information (name, period, spatial and temporal resolution, unit, type of data set (map, table), reference, etc.) on the different data sets that were compared might help a lot.

Such a table was added at the end of section 2.1 for the five data sets presented here. However, we did not include the spatial and temporal resolution because it is the same for all the data sets. This information is given in the text.

I cannot find a definition of the evaluation and projection run. It is not fully clear to mean what was done here.

Thank you for pointing out that this was not described clearly. The difference between the evaluation and the projection run is that the former was driven with reanalysis meteorological data (ERA40 and ERA5, based on observed data) while the projection run was driven with the output of a global climate model (MIROC5). We added the following sentences (line 183-187):
*"The evaluation simulation driven with reanalysis data serves as a reference for the historical simulation driven by a global climate model. It quantifies how well the historical climate can be reproduced by the historical simulation and how large the differences of specific climate variables are between both simulations. In addition, Rybka et al. (2022) used the evaluation simulation for a comparison with high resolution observational precipitation data sets to analyze the model performance for extreme precipitation."*

We hope that this makes the purpose of using an evaluation run clearer.

Can you provide a rough estimate on the sensitivity to use the KE-I relation by Wischmeier

and Smith 1978? There are many but some are more frequently used like the exponential function by Brown and Foster 1987 as it is suggested in the RUSLE by Renard et al. 1996.

Following the comment by Panos Panagos, we tested the effect of using the Brown and Foster (1987) equation on a subset of our data. In this case, the results obtained with the equation proposed by Wischmeier and Smith (1978) were on average 1.23 times higher than the ones obtained with the Brown and Foster equation.
Nearing et al., 2017 and Hanel et al., 2016 compared different KE-I equations.

This information was added in a new paragraph in the revised manuscript (line 323-347).

Here we did not test different KE-I equations systematically because of the considerable computation time to calculate R-factors for the large data set.

Results and discussions:

A brief discussion on the usability of CPS for landscape or field scale studies could be interesting. Just out of curiosity, how good does the CPS work for specific points in space? How good is a comparison against long term rainfall gauges?

We are not aware of any field scale studies comparing time series obtained from CPS to the data of specific rain gauges. Rybka et al. (2022) compared sub-daily extreme precipitation from COSMO-CLM to i) the German radar climatology dataset RADCLIM and ii) the KOSTRA-DWD product that statistically estimates extreme precipitation in Germany based on rain gauge observations. This comparison showed that the CPS performs well in reproducing observational data for durations above 12 h but overestimates hourly extreme precipitation intensities.

I do not understand the point of calibrating the transport capacity to end up with same soil redistribution rates. In section 2.3 - L218 the reason to apply WaTEM/SEDEM is named as "To study the effects of changing rainfall erosivity on soil erosion [...]". From my perspective the benefit to apply WaTEM/SEDEM is to get a rough number on the differences in soil redistribution and sediment delivery to the stream network in Mg ha-1 yr-1.

On the scale of Fig. 8, the differences between the realisations are not visible. From my perspective, Figure 9 provides a good relative number of the effect in R-factor calculation in CPS and MAE.

Following the recommendation of reviewer 2 and Shuiqing Yin, we deleted the part on soil erosion modelling to focus entirely on rainfall erosivity. Thus, the two figures were deleted and the calibration is not relevant. To respond to your comment nonetheless:

The transport capacity in WaTEM/SEDEM has to be calibrated to ensure that simulated sediment yields are as similar as possible to observed ones. Of course, this is only possible for the past. This was done for both R-Factor data sets (calculated from CPS and from MAP) so the simulated erosion rates shown in Fig. 8 are very similar. For the future projections on the other hand the changes in rainfall erosivity differ strongly between the two data sets (CPS vs. MAP) so the projected changes in rainfall erosivity shown in Fig. 9 are very different.

The drawback of using a single CPS instead of an ensemble is highlighted multiple times throughout the manuscript. Would it make sense to assess if the model tends to under or

overpredict rainfall erosivity in a comparison against rainfall ground observations (rain gauge or laser-distrometer)? See comment on comparison against rain gauge point data above.

We certainly agree that it is important to compare our results to rainfall erosivity calculated from observational data. Here we preferred using the R-factor maps published by other authors. We agree that a comparison to rain gauge point data would also be good. We will keep this in mind for future research.

Thank you for this nice piece of work!

Thanks for your appreciation of our work and thanks again for your time spent and your comments!

References

Brown, L. and Foster, C.: Storm erosivity using idealized intensity distributions. TRANS ASAE, 30(2), 378-386, 1987.

Hanel, M., Máca, P., Bašta, P., Vlnas, R., and Pech, P.: The rainfall erosivity factor in the Czech Republic and its uncertainty, HYDROL EARTH SYST SC, 20, 4307-4322, https://doi.org/10.5194/hess-20-4307-2016, 2016b.

Nearing, M. A., Yin, S.-q., Borrelli, P., and Polyakov, V. O.: Rainfall erosivity: An historical review, CATENA, 157, 357-362, https://doi.org/10.1016/j.catena.2017.06.004, 2017.

Rybka, H., Haller, M., Brienen, S., Brauch, J., Früh, B., Junghänel, T., Lengfeld, K., Walter, A., and Winterrath, T.: Convection-permitting climate simulations with COSMO-CLM for Germany: Analysis of present and future daily and sub-daily extreme precipitation, development, METEOROL Z 64, 65, https://doi.org/10.1127/metz/2022/1147, 2022.

Wischmeier, W. H. and Smith, D. D.: Predicting rainfall erosion losses-a guide to conservation planning, United States Department of Agriculture, Agriculture Handbook No. 537, 58 pp., 1978.

---

## Author Comment (AC5)

Dear Shuiqing Yin,

Thank you very much for your helpful comment and your interest in our study. Below, we answer to the points you raised one by one, marked in blue.

Best regards,

Magdalena Uber on behalf of all co-authors

The study proposed the advantage of Convection-Permitting Simulations (CPS) in simulating extreme convective precipitation, which plays an critical role in the soil erosion process. Uber et al. compared rainfall erosivity for the past (1971-2000), present (2001-2019), near future (2031-2060), and far future (2071-2100) periods in the Central European region based on CPS data as well as soil erosion in the Elbe River basin. The study follows in the scope of HESS and the use of CPS data for the estimation and projection of future erosivity is the novelty of the study. Following are the main concerned issues:

- Uber et al. used "CMIP5 driven CPS" to compare with "ERA5 driven CPS" for the evaluation simulation, which can not demonstrate the benefit of CPS in simulating extreme precipitation. High spatial-temporal resolutions of precipitation observations should be used for the purpose.

Apparently, we did not describe the purpose of the ERA5/ERA40 driven evaluation runs sufficiently. The comparison of the CMIP5 driven projection run with the ERA5/ERA40 driven evaluation run was not undertaken to demonstrate the benefit of CPS. Instead, the evaluation run serves as a reference that can be used for different applications: comparisons with other model simulations, with precipitation observations, driving data set for impact models.

We fully agree that CPS model results have to be compared to high resolution precipitation observations. This was done in a quality check routine before data publication and, in more detail, by Rybka et al. (2022). For rainfall erosivity, we compared our results to rainfall erosivity calculated by Auerswald et al., 2019 from high resolution radar precipitation data (Section 3.1.2).

In order to clarify the purpose of the evaluation run, we added the following sentences in the manuscript: "*The evaluation simulation driven with reanalysis data serves as a reference for the historical simulation driven by a global climate model. It quantifies how well the historical climate can be reproduced by the historical simulation and how large the differences of specific climate variables are between both simulations. In addition, Rybka et al. (2022) used the evaluation simulation for a comparison with high resolution observational precipitation data sets to analyze the model performance for extreme precipitation.*" (line 183-187).

- Regression equation based on mean annual precipitation sums can not be used to project future rainfall erosivity for it can not fully reflect the change of daily and hourly precipitation intensity along with the warming climate.

We certainly agree. This was also criticized by reviewer 2 so our answer is the same:

We argue that one of the main advances of using CPS for erosion modelling is that we don't have to rely on regression equations such as Equation 3 anymore. Nonetheless, we think that using Equation 3 also has advantages over using CPS, the main one being that it can easily be applied to (projected) climate data of basically any temporal resolution. In our case, it enables an estimate of changes in rainfall erosivity due to changes in mean annual precipitation from an ensemble of climate models. Thus, variability between climate models can be assessed which is not yet the case for CPS.

Apparently, we did not point this out clear enough in the manuscript. Thus, we changed Figure 6 so that it now includes results showing the bandwidth of the climate model ensemble. Furthermore, we added the following paragraph (line 377-392):

"Furthermore, the changes in rainfall erosivity calculated from convection-permitting climate model output are considerably higher than the ones calculated with the low-resolution approach using mean annual precipitation from model output of conventional regional climate model ensembles (Fig. 6). *This is the case not only when future MAP is obtained from the median of the model ensemble but also for the entire plausible bandwidth of models. Figure 6 shows changes in rainfall erosivity estimated with the 15th and the 85th percentile of the model ensemble. Even though this approach only considers changes in MAP and not changes in rainfall intensity, it allows an estimate of model uncertainty due to the differences between the ensemble members. The results obtained with CPS are outside of the bandwidth of the model ensemble because they also represent changes in extreme precipitation in addition to changes in MAP.*"

- This reviewer agrees with the RC2 that the temporal scaling factor of 1.9 is reasonable (Yue et al. (2020) as a reference, Effect of time resolution of rainfall measurements on the erosivity factor in the USLE in China) . Nearing et al. (2017, Rainfall erosivity: An historical review) reported the differences among three KE-I equations in USLE and its revisions, which may be useful for you to discuss differences between your study and Panagos et al. (2015c). Breakpoint or less than 5-min interval observation data are suggested to be obtained for the determination of the scaling factor of the study area.

Thank you very much for the useful references. We included them in the revised manuscript. The temporal scaling factor of 1.9 was obtained for Germany (i.e. the center of the study area) by Fischer et al., 2018 using 1 min resolution rain gauge data.

Of course, differences in KE-I equations are important. Besides Nearing et al., (2017), also Hanel et al. 2016 presented an interesting comparison of 14 different equations.

Following your comment and the one of reviewer 2, we added a paragraph that discusses these issues in the revised manuscript (line 323-347).

- This reviewer agrees with the RC2 that the chapter on future erosion rates for the Elbe River basin is strongly shorten or even deleted. Instead, the study focuses on the projection of rainfall erosivity and fully demonstrates the advantage of CPS data to highlight the novelty.

The parts on soil erosion modelling was deleted as suggested by you and reviewer 2.

References:

Hanel, M., Máca, P., Bašta, P., Vlnas, R., and Pech, P.: The rainfall erosivity factor in the Czech Republic and its uncertainty, HYDROL EARTH SYST SC, 20, 4307-4322, https://doi.org/10.5194/hess-20-4307-2016, 2016b.

Nearing, M. A., Yin, S.-q., Borrelli, P., and Polyakov, V. O.: Rainfall erosivity: An historical review, CATENA, 157, 357-362, https://doi.org/10.1016/j.catena.2017.06.004, 2017.

Rybka, H., Haller, M., Brienen, S., Brauch, J., Früh, B., Junghänel, T., Lengfeld, K., Walter, A., and Winterrath, T.: Convection-permitting climate simulations with COSMO-CLM for Germany: Analysis of present and future daily and sub-daily extreme precipitation, development, METEOROL Z 64, 65, https://doi.org/10.1127/metz/2022/1147, 2022.

---

## Referee Report (RR1)

The manuscript by Uber et al. about the study on using convection-permitting simulations (CPS) for estimation of future rainfall erosivity in Central Europe gained a lot by their revisions. The authors took carefully over several suggestions of the referees. By this, the manuscript improved, especially by removing the chapter on soil loss estimations and the focus on rainfall erosivity derived from the different precipitation data sets. The overview of the different data sets provided by table 1 is of benefit to understanding their descriptions. The added paragraph on the discussion of the correction factors is of high importance. Still, some minor revisions are suggested as follows:

The application of the regression in DIN 19708 for estimation of erosivity by using the mean annual precipitation (MAP) is still of critic. Your argument that the DIN 19708:2022 did not exist when you performed your calculations is not a good one as the DIN 19708:2002 should have already existed before your first submission of the manuscript. In any case, you bring in the arguments for no longer using the regression nicely yourself: I) In the lines 49 to 51 of the revised, tracked manuscript you write that 'low-resolution approaches' for estimation of future rainfall erosivity are not permitted as it is expected that the frequency distribution of rainfall events changes by climate change. II) In line 70 you are referring to studies which observed that rainfall erosivity already increased. III) In lines 83 to 85 you state that regression-based models using monthly or yearly sums of precipitation "are only valid for the time period for which these models are calibrated and lead to underestimations of the rainfall erosivity if extreme precipitation events increase with time, as suggested by many climate change scenarios.". IV) In lines 123 and 124 you are writing that using MAP for estimation of rainfall erosivity may not be valid for future climate with precipitation frequency and magnitude different from that of the precipitation used for establishing of the equation. In consequence of these notes (I to IV), it is surprising for the reader that you used the equation in DIN 19708. Therefore, it needs a clearer justification of the purpose of nevertheless using this equation. Chapter 2.2.2 provides still just the explanation that you used the low temporal resolution approach "for comparison". This seems not enough to understand the purpose of using the equation. This is still the case albeit inserting "15th and 85th percentile" in line 237 of the revised, tracked manuscript. I highly recommend clarifying earlier than in the results & discussion part (lines 388 to 392) the actual purpose and chance of including the low-resolution approach. In addition, I suggest referring that the current version of DIN 19708 (DIN19708:2022) explicitly states that the regression based on MAP can only be used for 'historical observations'.

The added section on the comparison of erosivities calculated by using 'USLE-based' and the 'RUSLE-based' equation is very interesting. Still, I suggest to remove or revise the detailed discussion of the discrepancy of your results and the results of Panagos et al. (lines 344-347). There might be more differences between your approach and the one of Panagos et al. than you mention there, e. g. the spatial and temporal coverage as well as the different criteria for defining erosive rainfall events. The criteria are of importance for the number of events. Lower thresholds for the maximum 30 min rain intensity result in a higher number of erosive events which sum up to higher annual erosivity. Your criteria might deviate from the ones used by Panagos et al. in the study to which you are referring to. Therefore, I suggest to either complete the detailed discussion of explanations for the discrepancies between the two studies by considering all possible causes, or to delete it (lines 344 - 347) and just discuss consequences of the differences between USLE-based and RUSLE-based R factors.

The chapter 3.1.3 is part of your results and discussions chapter but misses a more intensive discussion of possible reasons for the discrepancies of your results to the studies you are referring to. It would be of interest I) whether the scatter of the erosion index increased from past to near to far feature as one would expect from a possible increase in extreme events occurring on single days in limited areas, and II) what the reasons could be for your results not showing seasonal changes of the erosion index.

Instead of deleting the sentence "Thus, there might be regions in which seasonal shifts occur, but that average out over the larger modelling domain used in this study", I encourage you to discuss it and to clearly state that this is an open question which you cannot answer so far, if this is the case.

Some more minor comments referring to the revised, tracked manuscript are following:

Page 5, line 142:         I suggest to already state the applied emission scenario (RCP8.5).

Page 11, lines 290-294:  This paragraph doesn't seem to fit here. Might be moved to the introduction.

Page 12, line 343:        The wording "Our values" is confusing; maybe you can rephrase it to "The USLE-based R factors".

Page 16, line 419/420:  Do you mean with "…the trends in the two data sets can differ considerably" that the trends differ in some regions of your spatial extent? Moreover, what do you mean with "usually" in the following sentence? Do you mean in most of the area?

---

## Author Response (AR2)

Dear referee,

Thank you very much for appreciation of our revisions, for your repeated effort to review the manuscript and your further helpful comments.

In the following, your comments are written in black and our responses in blue. Changes made in the manuscript are indicated with Italics.

Best regards,

Magdalena Uber on behalf of all co-authors

The manuscript by Uber et al. about the study on using convection-permitting simulations (CPS) for estimation of future rainfall erosivity in Central Europe gained a lot by their revisions. The authors took carefully over several suggestions of the referees. By this, the manuscript improved, especially by removing the chapter on soil loss estimations and the focus on rainfall erosivity derived from the different precipitation data sets. The overview of the different data sets provided by table 1 is of benefit to understanding their descriptions. The added paragraph on the discussion of the correction factors is of high importance. Still, some minor revisions are suggested as follows:

The application of the regression in DIN 19708 for estimation of erosivity by using the mean annual precipitation (MAP) is still of critic. Your argument that the DIN 19708:2022 did not exist when you performed your calculations is not a good one as the DIN 19708:2002 should have already existed before your first submission of the manuscript. In any case, you bring in the arguments for no longer using the regression nicely yourself: I) In the lines 49 to 51 of the revised, tracked manuscript you write that 'low-resolution approaches' for estimation of future rainfall erosivity are not permitted as it is expected that the frequency distribution of rainfall events changes by climate change. II) In line 70 you are referring to studies which observed that rainfall erosivity already increased. III) In lines 83 to 85 you state that regression-based models using monthly or yearly sums of precipitation "are only valid for the time period for which these models are calibrated and lead to underestimations of the rainfall erosivity if extreme precipitation events increase with time, as suggested by many climate change scenarios.". IV) In lines 123 and 124 you are writing that using MAP for estimation of rainfall erosivity may not be valid for future climate with precipitation frequency and magnitude different from that of the precipitation used for establishing of the equation. In consequence of these notes (I to IV), it is surprising for the reader that you used the equation in DIN 19708. Therefore, it needs a clearer justification of the purpose of nevertheless using this equation. Chapter 2.2.2 provides still just the explanation that you used the low temporal resolution approach "for comparison". This seems not enough to understand the purpose of using the equation. This is still the case albeit inserting "15th and 85th percentile" in line 237 of the revised, tracked manuscript. I highly recommend clarifying earlier than in the results & discussion part (lines 388 to 392) the actual purpose and chance of including the low-resolution approach. In addition, I suggest referring that the current version of DIN 19708 (DIN19708:2022) explicitly states that the regression based on MAP can only be used for 'historical observations'.

We understand your criticism and are convinced that the use of CPS is a good way to overcome the limitations of using MAP. To stress this, but also to explain why we used the approach nonetheless we added the following lines in the manuscript, section 2.2.2:

*"The low temporal resolution approach was used here because it allows a representation of the bandwidth of results obtained with a regional climate model ensemble and thus an estimate of model uncertainty which is not yet possible for CPS. Nonetheless, the main limitation, i.e. the neglection of the effect of increases in heavy rain, of the approach has to be stressed again. This shortcoming is overcome by CPS and is one of the reasons why the most recent version of DIN 19708 (DIN 19708:2022-08) recommends to use equation 3 only for historical observations"* (line 240-244 in the tracked manuscript in second revision)

The added section on the comparison of erosivities calculated by using 'USLE-based' and the 'RUSLE- based' equation is very interesting. Still, I suggest to remove or revise the detailed discussion of the discrepancy of your results and the results of Panagos et al. (lines 344-347). There might be more differences between your approach and the one of Panagos et al. than you mention there, e. g. the spatial and temporal coverage as well as the different criteria for defining erosive rainfall events. The criteria are of importance for the number of events. Lower thresholds for the maximum 30 min rain intensity result in a higher number of erosive events which sum up to higher annual erosivity. Your criteria might deviate from the ones used by Panagos et al. in the study to which you are referring to. Therefore, I suggest to either complete the detailed discussion of explanations for the discrepancies between the two studies by considering all possible causes, or to delete it (lines 344 - 347) and just discuss consequences of the differences between USLE-based and RUSLE-based R factors.

Thank you for this comment. We agree. As we cannot quantify all possible causes for the discrepancies, we deleted the respective lines as you recommended.

The chapter 3.1.3 is part of your results and discussions chapter but misses a more intensive discussion of possible reasons for the discrepancies of your results to the studies you are referring to. It would be of interest I) whether the scatter of the erosion index increased from past to near to far feature as one would expect from a possible increase in extreme events occurring on single days in limited areas, and II) what the reasons could be for your results not showing seasonal changes of the erosion index. Instead of deleting the sentence "Thus, there might be regions in which seasonal shifts occur, but that average out over the larger modelling domain used in this study", I encourage you to discuss it and to clearly state that this is an open question which you cannot answer so far, if this is the case.

Thanks again for the comment. We will keep your remark I) in mind for further analysis. Concerning your remark ii) we actually do not know what are the reasons for the discrepancies, so we stated this in the manuscript as you suggested by adding *"The reasons for the discrepancies between this study which did not detect significant changes in the seasonal distribution and the other studies that did observe trends are not clear yet and an remain an open question."* in line 367-369.

Some more minor comments referring to the revised, tracked manuscript are following:

Page 5, line 142: I suggest to already state the applied emission scenario (RCP8.5).

It was added as you suggested.

Page 11, lines 290-294: This paragraph doesn't seem to fit here. Might be moved to the introduction.

We prefer to leave this paragraph here in order not to overcharge the introduction. It is supposed to show a further possible use of the data that is presented in this section. To make this clearer we changed the sentence slightly:

"As such, *the rainfall erosivity data presented here* can also provide valuable information for other hydrological applications dealing with extreme rainfalls such as the assessment of (future) risks for flash floods or landslides or identifying zones that are prone to these natural risks (Fiener et al., 2013; Panagos et al., 2015c)." (line 296-298)

Page 12, line 343: The wording "Our values" is confusing; maybe you can rephrase it to "The USLE-based R factors".

It was changed as suggested (line 348).

Page 16, line 419/420: Do you mean with "…the trends in the two data sets can differ considerably" that the trends differ in some regions of your spatial extent? Moreover, what do you mean with "usually" in the following sentence? Do you mean in most of the area?

This refers to the comparison of the other studies presented in table S1 to our values. Using the data from the evaluation run differs from using the data of the simulation run with stronger trends being found in the simulation run in most cases. As the time period for these cases is always the same, "usually" refers to most of the considered regions. Thus, "usually" was replaced by "*in most regions*" in line 424.